# KRT17 promotes triple negative breast cancer through activation of Wnt signaling and γδ T-cells recruitment
Chermakani Panneer Selvam[1,2], Gatha Thacker [1,2], Ukjin Kim[1,2], Youley Tjendra[3], Melinda M. Boone[1], Samantha Henry[4], Camila O. Dos Santos[4] & Rumela Chakrabarti [1,2] ✉

Triple-negative breast cancer (TNBC) is a very aggressive form of breast cancer and Black American (BA) women face disproportionately higher mortality rates than White American (WA) women. The molecular mechanism behind this disparate clinical outcome remains poorly understood. We find that BA TNBC patients exhibit higher protein expression of KRT17 compared to WA TNBC and non-TNBC patients and correlates to poor distant metastasis-free survival. Mechanistic studies in metastatic mouse TNBC tumors with higher Krt17 demonstrates higher Wnt signaling targets, cancer stem cells (CSCs), which positively correlates to several metastasis signatures, supporting clinical data. Consistently, KRT17[high] BA patient tumors display higher activated Wnt signaling. Furthermore, Krt17 regulates Wnt signaling to drive recruitment of γδ T-cells in both mouse and human samples, which can be reversed by targeting Wnt signaling, identifying the Krt17-Wnt signaling axis as a critical driver of clinical disparities and a novel targetable vulnerability for BA TNBC patients.

Breast cancer is the leading cause of cancer related death in women. Triple-negative breast cancer (TNBC) is a subtype of breast cancer that accounts for 15-20% of all breast cancer, through lack of estrogen receptor, progesterone receptor, and human epidermal growth factor receptor 2 (ER⁻/PR⁻/HER2⁻)[1]. TNBC is responsible for 35% of breast cancer related deaths. Molecular studies have demonstrated that TNBC encompasses as heterogeneous disease, with diverse subtypes with distinct transcriptomic profile[2,3]. In TNBC, heterogeneity has been connected with poor prognosis, higher risk of distant metastasis, and high recurrence rate than other types[4,5]. TNBC also accounts for increased mortality in BA TNBC women[6–8]. The molecular mechanism behind this disparity remains poorly understood. At present, conventional chemotherapy and radiation are the standard backbones of the treatment of TNBC patients; however, better stratification of TNBC patients is warranted for better targeted treatment options.

Keratin 17 (Krt17) is a type-I intermediate filament protein in respiratory epithelium, urothelium, and various glands such as sebaceous, salivary, and eccrine sweat glands, and Krt17 is regarded as a basal/myoepithelial cell keratin in breast based on localization[9–12]. Some conflicting reports mention the correlation of Krt17 to breast cancer[13,14], limiting the comprehensive understanding of Krt17 in breast cancer progression and metastasis. The overexpression of Krt17 was also noted in the physiological

and pathological condition like, wound healing process in the early stage of epidermal stressors or injury[15]. Furthermore, Krt17 can act as an inflammation and immunity modulator, as seen by studies showing overexpression of Krt17 altered T-cells immune responses in skin diseases such as psoriasis[16,17]. In addition, Wang et al. reported that the overexpression of Krt17 is involved in immune evasion and resistance to the immune checkpoint blockade in malignant tumors[18]. Other cancer studies, such as esophageal squamous cell carcinoma, ovarian, and cervical cancer, also showed that high level expression of Krt17 is associated with shorter survival of patients[19–21].

In 2022, Wang et al.[22] reported that overexpression of Krt17 regulates proliferation and invasiveness in laryngeal squamous cell carcinoma through Wnt/β-catenin signaling pathway. In normal mammary gland development, Wnt signaling plays a major role during embryogenesis, and dysregulation of Wnt signaling in breast cancer cells promotes breast cancer initiation and aggressiveness, particularly in TNBC[23–26]. However, what regulates Wnt signaling in diverse TNBC patient subsets is not well understood. Earlier studies clearly indicate that Wnt signaling is extensively involved in T-cell development, differentiation, and function[27–29]. However, the mechanistic connection of Krt17 and Wnt signaling and immune cells in breast cancer has not yet been explored, particularly in the context of disparate clinical outcomes of patients.

[1]Sylvester Comprehensive Cancer Center, University of Miami, Miami, FL, USA. [2]Department of Surgery, Miller School of Medicine, University of Miami, Miami, FL, USA. [3]Department of Pathology and Laboratory Medicine, Miller School of Medicine, University of Miami, Miami, FL, USA. [4]Cold Spring Harbor Laboratory, Cold Spring Harbor, NY, USA. ✉e-mail: rxc1335@miami.edu

Here, we demonstrate that KRT17 expression is higher in aggressive TNBC tumors, including in BA TNBC patients, compared to WA TNBC and non-TNBC patients. High *KRT17* expressing BA TNBC patients have poor Distance Metastasis-Free Survival (referred hereafter as DMFS) compared to WA TNBC patients, suggesting that high level of *KRT17* could be used as a predictive marker for poor clinical outcome in BA TNBC patients. Single cell RNA-sequencing (scRNA-seq) analysis in representative clinically relevant spontaneously metastatic mouse TNBC tumors identified positive correlation of high levels of *Krt17* tumor with increased Wnt signaling. Functional studies demonstrated knockdown of Krt17 (Krt17 KD) is associated with reduced tumor metastasis through reduced cancer stem cells (CSCs) depending on β-catenin dependent Wnt signaling and γδ T-cells immune cells population in the tumor microenvironment (TME) of TNBC. Consistently, higher population of γδ T-cells and upregulation of Wnt signaling is observed more in *KRT17*[high] BA TNBC patient's samples compared to WA counterparts. Together, our studies identify the novel tumor promoting function of KRT17 in BA TNBC through Wnt signaling and γδ T-cells recruitment, which could be used to predict poor clinical outcome in BA TNBC patients. Preclinical studies show that pharmacological inhibition of Wnt signaling could open new vulnerabilities in BA TNBC patients with higher KRT17 expression.

## Results

### Higher levels of KRT17 correlate with poor prognosis in BA TNBC patients

To estimate the prognostic information of Krt17 in different subsets of breast cancer, which have higher *KRT17* (Fig. 1a–c), we investigated the correlation of *KRT17* expression in different breast cancer subtypes using Kaplan-Meier (KM) plotter[30]. KM survival analysis showed that TNBC patients with high *KRT17* gene expression indicate a shorter Distance Metastasis Free Survival (referred hereafter as DMFS) (Fig. 1a–c) (TNBC, $P = 0.022$) compared to luminal A and B subtypes ($P = 0.11$ and $P = 0.093$), suggesting that *KRT17* may be a marker for poor prognosis specifically in TNBC subset. We found that TNBC subset of breast cancer had highest expression of *KRT17* compared to other subtypes (Fig. 1d) using database in-silico analyses from UALCAN platform (http://ualcan.path.uab.edu)[31,32]. TNBCs are very heterogeneous[33], so we next examined the correlation of *KRT17* in different TNBC subsets and found that patients with high *KRT17* gene expression have significantly shorter DMFS in immunomodulatory (correlated to more immune cell infiltration and immune suppressive tumor micronevironment) subtypes and mesenchymal (connected to more stem-like features) subtypes of TNBC ($P = 0.029$ and $0.0065$, respectively, Fig. 1e, f). Both these subsets of TNBC are known to have poor clinical outcome such as higher metastasis and increased resistance to standard therapy. Lymph node metastasis is an independent prognostic factor in the early stage of TNBC[34]. The DMFS of TNBC patients with lymph node metastasis (Fig. 1g) with high-level expression of *KRT17* is lower than TNBC patients with low level expression of *KRT17* ($P = 0.08$), suggesting that over-expression of KRT17 is associated with higher lymph node metastasis, corroborating earlier data.

To determine if similar correlation of KRT17 and patient outcome exists at protein levels, we performed immunohistochemistry with 38 TNBC patients and examined the KRT17 protein expression and found that TNBC ($n = 17$) patient samples (Supplementary Table 1) showed a higher level of KRT17 protein expression ($P = 0.0005$) as compared with the non-TNBC ($n = 21$) (Fig. 1h, i), corroborating with our in-silico RNA data. We next tested the correlation of high protein level of KRT17 with survival using another cohort of TNBC patient tissues. Good survival was defined as (≥6 years survival) and poor survival (less than 6 years survival) in TNBC patients[35]. As shown in Fig. 1j, k, poor survival ($n = 9$) TNBC patients have more KRT17 protein expression than good survival ($n = 7$) TNBC patients (Supplementary Table 2).

In USA, TNBC has a 2-fold worse poor survival rate in BA than WA patients[6]. So, we next tested KRT17 protein expression in two separate TNBC patient cohorts with specific race information, WA

($n = 29$) and BA ($n = 29$) TNBC (Supplementary Table 3). We found that BA TNBC patient's tissues express a higher level of KRT17 protein as compared to the WA (Fig. 1l, m). Normal adjacent tissues (NAT) from BA and WA TNBC patient samples (Supplementary Table 4) were also tested for KRT17 protein levels, and we found that expression of KRT17 were higher in tumor tissues than normal tissues from both races (in Supplementary Fig. 1a, b). Also, in normal tissues, KRT17 was primarily expressed in basal/myeoepithelial cells, whereas in cancer tissues, KRT17 expression was present in most areas. Due to low number of normal adjacent tissues and different expression pattern, we were unable to quantify the difference. To determine the level of KRT17 in TNBC tumors from other races, we also obtained formalin fixed slides from Asian TNBC patients ($n = 105$, Tissue microarray from Creative Bioarray) and found that BA TNBC patients have higher trend for expression of KRT17 than Asian TNBC patients ($P = 0.2610$, Supplementary Fig. 1c, d). WA TNBC patients had lower expression of KRT17 than BA TNBC patients, $P = 0.0174$, Supplementary Fig. 1c, d). This data was also corroborated using UALCAN platform, where we found that BA TNBC patients having higher KRT17 protein levels compared to WA TNBC patients (Supplementary Fig. 1e). *KRT17* was seen to be amplified in several breast cancer datasets (supplementary Fig. 1f). We next determined correlation between BA and WA TNBC patient survival and *KRT17*[high] expression in TNBC patients from the TCGA database using the UALCAN platform (http://ualcan.path.uab.edu)[31,32], since KM plotter did not have the race information to perform this analysis. Survival probability curves showed that BA TNBC patients with *KRT17*[high] expression had poorer overall survival rates in 1000 and 2000 days ($P = 0.029$) (Fig. 1n) compared to WA TNBC patients. Overall, these data indicate that *KRT17* may serve as a prospective poor prognostic biomarker and poor clinical outcome in TNBC patients in the USA.

### scRNA-sequencing of spontaneous metastatic mouse TNBC tumors identifies *Krt17*[+] cell cluster with enriched Wnt signaling

To test the function of Krt17 in TNBC, we next examined published scRNA-seq dataset of a novel metastatic spontaneous TNBC mice model (Supplementary Fig. 2a), which was recently developed by our laboratory[35,36] to investigate the molecular mechanism driving metastasis in TNBC patients. Notably, this model mimics TNBC patients as demonstrated by the correlation of gene signatures in an earlier study[37]. As discussed in recent publication[35,36], scRNA-seq data from metastatic TNBC (C3-T[+]; Elf5[+/−], called C3-T HET) and low metastatic TNBC (C3-T[+]; Elf5[+/+], called C3-T WT) mice models revealed 10 cell clusters in CD45 negative (CD45[−], also shown as CD45_N) population. During initial cluster analysis of tumor cells (CD45[−]), we found that *Krt17* is mostly expressed equally in the six epithelial clusters shown by high *EpCAM* expression (0, 1, 2, 3, 4, and 9) and not in the four fibroblasts clusters shown by high *Sparc* expression (5, 6, 7, and 8) (Fig. 2a–d and Supplementary Fig 2b). We next investigated the properties of *Krt17*[+] vs *Krt17* tumor cells using Gene Set Enrichment Analysis (GSEA). Krt17 > 0 is considered as Krt17[+] cells. Gene signatures indicated that *Krt17*[+] TNBC tumor cells were highly connected to metastasis, stemness, hypoxia, *Brca1* mutation, glycolysis, and cysteine and methionine metabolic gene signatures, suggesting a connection of Krt17 to aggressive features of disease in TNBC (Fig. 2e–l). We next examined gene signatures of *Krt17*[+] cells from highly metastatic (C3-T HET) and low metastatic (C3-T WT) mouse TNBC tumors[35] and found that several signatures associated with tumor progression, such as hypoxia and several β-catenin dependent activated Wnt signaling were upregulated in metastatic (C3-T HET) tumor cells compared to low metastatic (C3-T WT) tumor cells (Fig. 2m–o), suggesting a connection of Krt17 to activated Wnt signaling in metastatic TNBC. High *Krt17* expression was observed in sorted CSCs (CD24[-]CD44[+] population was used[38]) from C3-T HET tumors compared to C3-T WT tumors (Fig. 2p), suggesting that Krt17 may play an important role in CSCs function in metastatic TNBC.

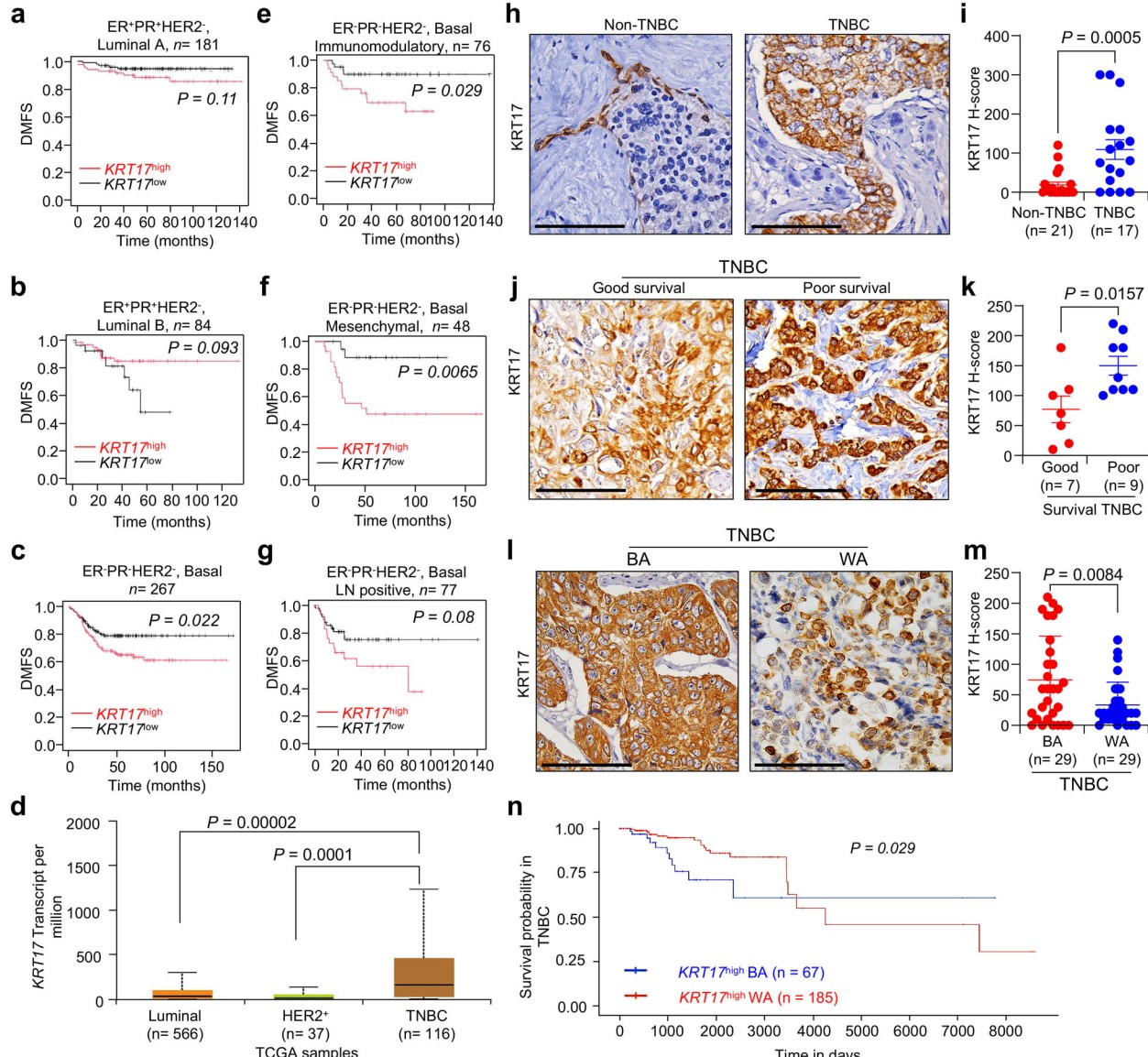

**Fig. 1 | Krt17 is higher in BA TNBC patients and correlates to poor clinical outcome. a, b** Distance metastasis free survival (DMFS) by the expression levels of *KRT17* from non-TNBC patients subtype (Luminal A, *n* = 181, Luminal B, *n* = 84) using Kaplan–Meier estimation database. **c** DFMS showing *KRT17* from TNBC basal subsets (*n* = 267). **d** Expression of *KRT17* in breast cancer subtypes using UALCAN database. **e, f** DMFS of *KRT17* high and low TNBC patients in different subset of TNBC immunomodulatory (*n* = 76) and mesenchymal (*n* = 48). **g** DMFS showing *KRT17*[high] and *KRT17*[low] TNBC patients with lymph node metastasis (*n* = 77). **h, i** Representative IHC staining and quantification showing protein expression level of KRT17 in tumors from TNBC (*n* = 17) and non-TNBC patients (*n* = 21). **j, k** Representative IHC staining of KRT17 protein expression and quantification in tumors from TNBC good (*n* = 7) and poor survival (*n* = 9) patient tissues, shows higher expression in poor survival patients. Within TNBC patients, poor survival was defined as less than 6 years of survival, and good survival as more than 6 years[35]. **l, m** Representative IHC staining of KRT17 protein expression and quantification in in BA (*n* = 29) and WA (*n* = 29) patient's tissues, shows higher expression in BA TNBC patients. **n** Survival probability of *KRT17* high BA and WA TNBC patients using UALCAN database, showed poor survival in BA TNBC patients (blue line). Data are presented as means ± SEM (Standard error of the mean). The two tailed Student's *t*-test was used to analyze statistically significant differences. The KRT17 H-Score was determined by multiplication of intensity and abundance of cells. Scale bars, 100 μm.

## Krt17 promotes cancer stem cells (CSCs) and TNBC tumor progression and metastasis

To examine the function of Krt17 in TNBC cells, we stably knocked down Krt17 (Krt17 KD) in 4T1 TNBC[39,40] cells, which mimic human TNBC and are a good metastasis model, using a shRNA lentivirus (Fig. 3a, b). The colony forming ability assay showed that Krt17 KD in 4T1 cells significantly reduced the colony forming ability as compared to the control group (Fig. 3c), suggesting reduced proliferation with reduced levels of Krt17. Next, we evaluated the CSCs property of the Krt17 KD 4T1 cells using the tumorsphere assay, a surrogate in vitro assay for CSCs. CSCs are a predominant factor for the distant metastasis and recurrence of breast

cancer[41,42]. We found that Krt17 KD TNBC cells show decreased tumorsphere formation (Fig. 3d–f), supporting earlier data. Also, we observed a significant decrease in the CSCs population in the Krt17 KD 4T1 cells (Fig. 3g) compared to control by FACS analysis. This data indicated that Krt17 plays a role in CSC number and function. Next, we performed Krt17 KD in non-TNBC cell line, E0771 (luminal B breast cancer subset)[43] to test if Krt17 promotes function of CSC in TNBC subset predominantly. Krt17 KD in E0771 cells did not show any change in the colony forming (Fig. 3h–j). There are no significant changes in the number of tumorsphere from E0771 cells, but we noted that the size of the tumorsphere was increased in Krt17 KD E0771 cells (Fig. 3k–m), corroborating earlier clinical data

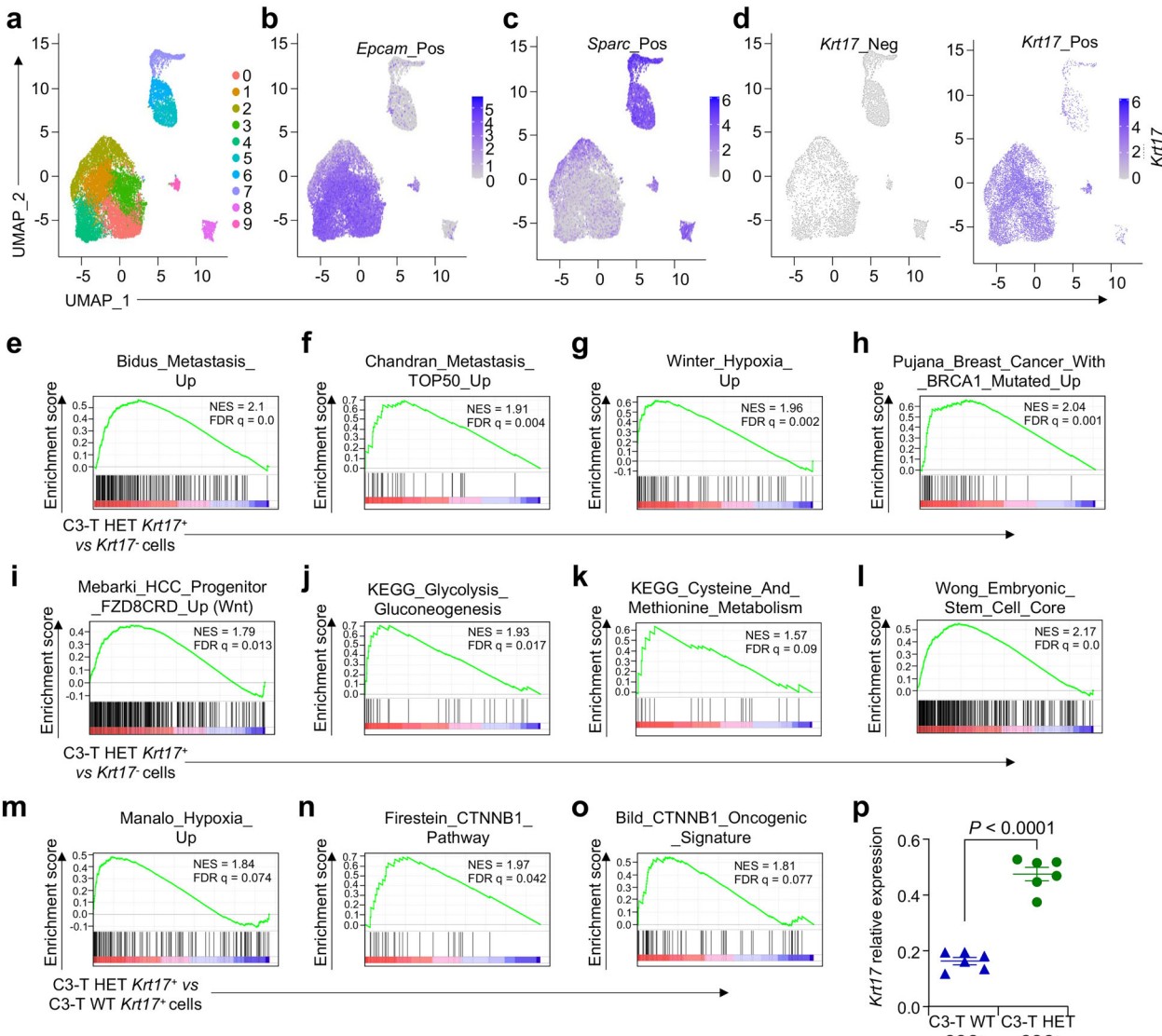

**Fig. 2 | scRNA sequencing identified CSCs in clinically relevant TNBC mouse model to have higher Krt17 and the Krt17⁺ cells have high Wnt signaling genes.** **a** Uniform manifold approximation and projection (UMAP) depicts the combined C3-T HET and WT CD45⁻ cells clusters. **b, c** UMAP visualization of EpCAM⁺ tumor epithelial cells and Sparc⁺ fibroblasts in C3-T HET and WT respectively. **d** UMAP shows distribution of Krt17⁺ cells population in epithelial clusters in C3-T HET tumors. **e–l** Gene set enrichment analysis (GSEA) from scRNA-seq shows higher stem cells, metastasis, hypoxia, BRCA1 mutation, metabolism, and Fzd8 signaling signatures in C3-T HET *Krt17*⁺ cells as compared to the C3-T HET *Krt17*⁻ cells. **m–o**

Gene set enrichment analysis (GSEA) shows higher Wnt/β-catenin signaling, and hypoxia signatures in C3-T HET *Krt17*⁺ cells as compared to the C3-T WT *Krt17*⁺ cells (NES, normalized enrichment score; FDR, false discovery rate). **p** C3-T HET sorted cancer stem cells (CSCs) (*n* = 3 individual tumors/group with technical duplicates) from indicated tumors show a higher expression level of *Krt17* as compared to the C3-T WT CSCs by qRT-PCR. *Krt17* expression values were normalized with *Gapdh*. Data are presented as the mean ± SEM. Statistical significance was determined by the Student's *t*-test.

(Fig. 1b), where we demonstrated that KRT17^high breast cancer patients are correlated to different outcomes in different subsets. This data indicates a functional role of Krt17 in breast cancer and highlights TNBC specific function of Krt17.

To strengthen our above in vitro findings, we evaluated the function of Krt17 in in vivo TNBC models. We injected control and Krt17 KD 4T1 TNBC cells into the mammary fat pad (MFP) of Balb/c mice and measured tumor progression. Krt17 KD cells showed slower tumor growth compared to control (Fig. 4a, b). Furthermore, compared to animals injected with control cells, mice injected with Krt17 KD 4T1 cells had significantly decreased lung metastasis (Fig. 4c, d). Lymph node metastasis is an early sign of disease spreading in TNBC[34]. Our immunofluorescence analysis in the lymph nodes of experimental mice revealed a decrease in number of Krt14⁺ single tumor cell metastasis in lymph nodes of Krt17 KD 4T1 tumor bearing

mice, similar to primary tumors in KD groups in comparison to the control group (Fig. 4e–h), supporting clinical data showing connection of Krt17 to metastases.

Furthermore, to validate the function of Krt17 in TNBC tumor progression, we injected C3-T HET cells (primary TNBC cells) into C3(1)/Tag-REAR (hereafter termed as Rear) mice, which contain all immune cells, but are more tolerant to cancer carrying C3-T antigen[44]. Enrichment of tumor epithelial cells from C3-T HET tumors was obtained using EpCAM⁺ cells enrichment microbead kit, followed by mammary fat pad injection into Rear mice[35,36]. Interestingly, KD of Krt17 in primary EpCAM⁺ C3-T HET cells leads to reduced tumor progression (Fig. 4i–k) and reduced metastasis in the lymph nodes (Fig. 4l, m) compared to control. Altogether, these results demonstrate that Krt17 is important for tumor progression and metastasis in multiple clinically relevant TNBC mice models.

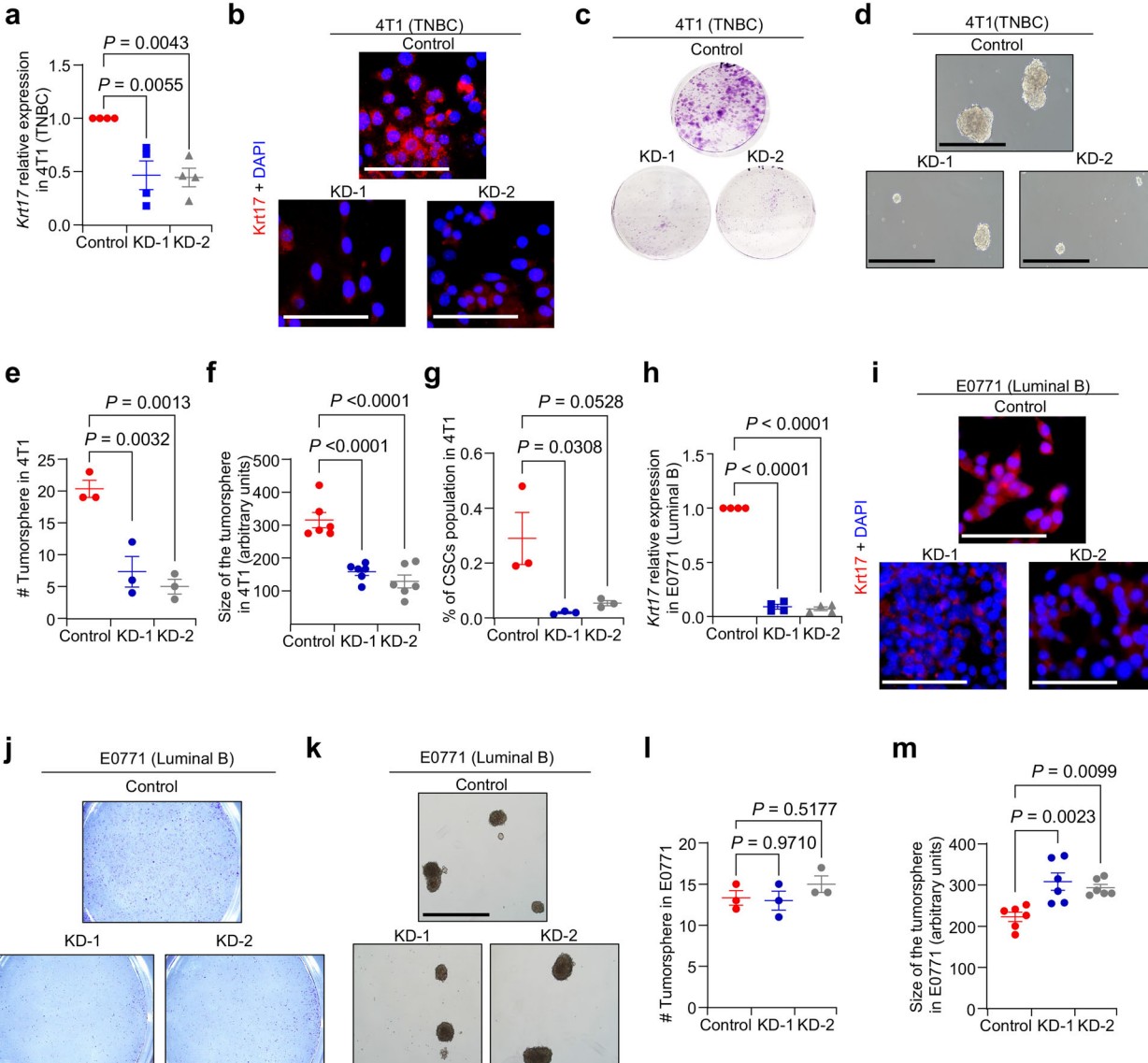

**Fig. 3 | Knockdown of Krt17 reduces CSCs in TNBC subset. a** qRT-PCR shows reduced *Krt17* mRNA levels in KD of Krt17 4T1 cells. Two biological samples were used with technical duplicates, *Krt17* expression values were normalized with *Gapdh*. **b** Immunofluorescence (IF) shows the reduced expression of KRT17 protein expression in Krt17 KD 4T1 TNBC cells compared to control (FOV *n* = 5 random field/samples). **c** Colony forming ability of KD of Krt17 in 4T1 cells (*n* = 3 independent experiments were performed/group). **d** Representative images and **e, f** Tumorsphere assay show that reduced sphere numbers and smaller sphere size from Krt17 KD 4T1 cells (*n* = 3 independent experiments/group, 2 spheres per group were evaluated for size estimation). **g** Scatter blot shows reduced percentage (%) of CSCs population in 4T1 cells from different groups (*n* = 3 samples/group).

**h** qRT-PCR shows reduced *Krt17* mRNA levels in Krt17 KD E0771 cells compared to control. Two biological samples were used with technical duplicates; *Krt17* expression values were normalized with *Gapdh*. **i** Immunofluorescence (IF) shows the expression Krt17 protein expression in non-TNBC cells (FOV *n* = 5 random field/group). **j** Colony forming ability of control and KD of Krt17 E0771 cells (*n* = 2 independent experiments). **k–m** Tumorsphere number (*n* = 3 independent experiments/group, 2 spheres per group was evaluated for size estimation) shows no changes in number of spheres from Krt17 KD E0771 cells compared to control cells. Data are presented as the mean ± SEM. Statistical significance was determined by one-way ANOVA with Tukey's multiple comparisons test. Scale bars, 100 μm.

## Krt17 regulates Wnt signaling dependent CSCs and abundance of γδ T-cells in TNBC tumor microenvironment (TME)

Activation of Wnt signaling is a key factor in breast cancer progression and metastasis[45]. Our GSEA signature results suggest aberrant activation of Wnt signaling in metastatic C3-T HET *Krt17*[+] cells compared to the C3-T WT *Krt17*[+] cells (Fig. 2n, o). Therefore, we next examined if Krt17 regulates Wnt signaling. Krt17 KD 4T1 cells show reduced levels of Wnt target genes such as *Lef1, Tcf1,* and *Ascl2*, as compared to the control 4T1 cells (Fig. 5a–c), suggesting that Krt17 may regulate Wnt signaling. Furthermore, we used lentiviral based Wnt reporter 7xTcf-eGFP reporter (7TGC)[46] to examine if Wnt signaling in TNBC cells is regulated by Krt17. As shown in Fig. 5d, we generated stably expressing 7TGC

reporter in control and Krt17 KD 4T1 cells. 7TGC lentivirus contains GFP cassette (readout of reporter) under the activation of seven Tcf-binding sites, and mCherry expression under the constitutively active SV40 promoter to show efficiency of infection[35,46]. Thus, mCherry fluorescence signal was a readout of infection efficiency, whereas GFP color is readout of Wnt reporter (Fig. 5e). Interestingly, transduction efficiency was higher in Krt17 KD1 4T1 cells as seen by higher mCherry expression, however, KD of Krt17 in 4T1 cells (both KD1 and KD2) significantly decreased the Wnt reporter GFP[+] fluorescence signal as compared to the control (Fig. 5f), indicating reduced Wnt reporter activity with loss of Krt17. Overall, these observations show that Krt17 in tumor cells is responsible for Wnt signaling activation.

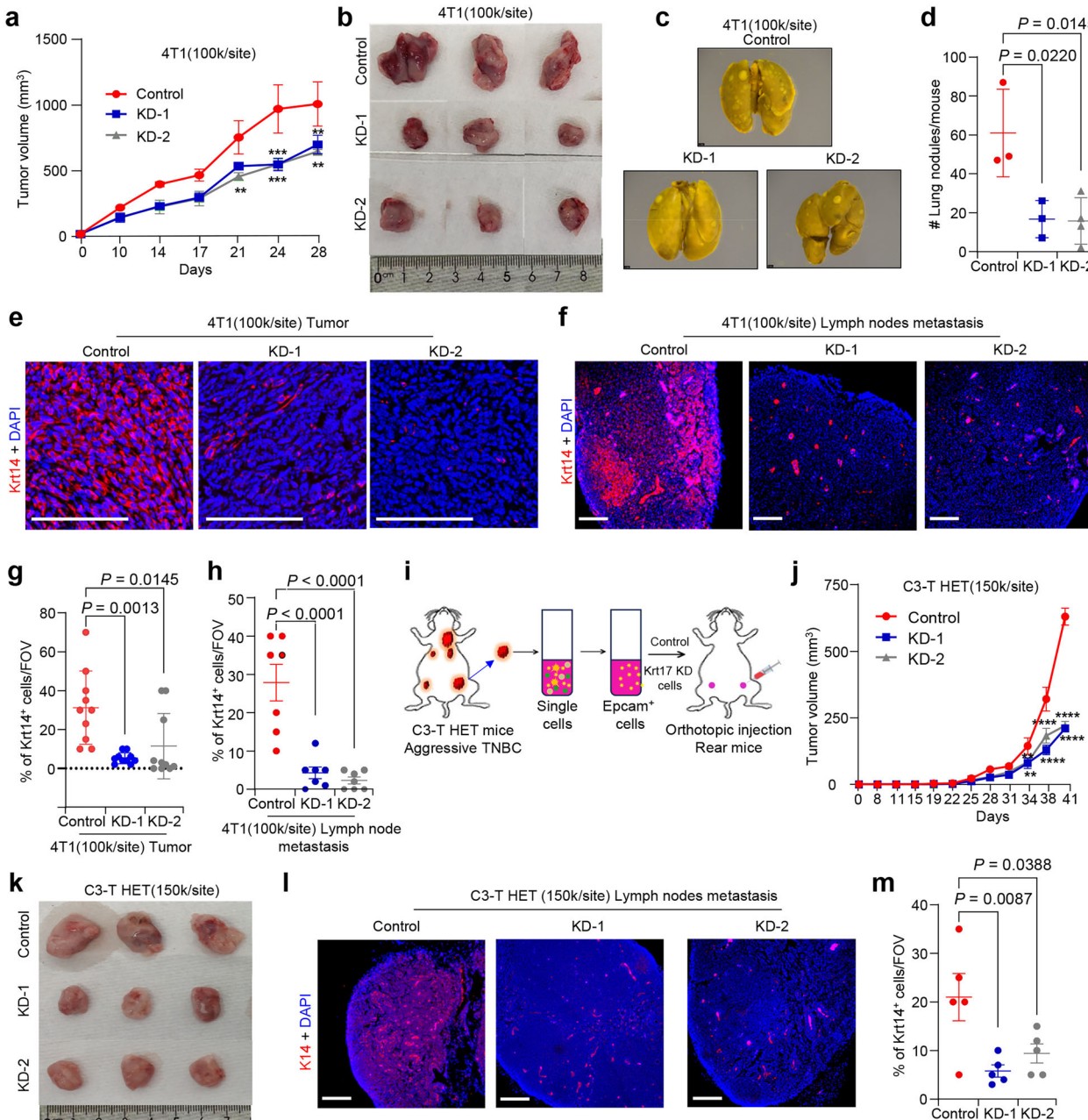

**Fig. 4 | Knockdown of Krt17 reduced TNBC progression and metastasis.**
**a**, **b** Tumor progression is reduced in Krt17 KD 4T1 TNBC cells compared to control. **c**, **d** Representative lung metastatic nodules images and quantification show reduced lung metastasis in Krt17 KD 4T1 cells injected mice ($n = 4$ tumors/group, 2 independent biological experiments were carried out). **e–h** Representative images and quantification Krt14 in primary tumors and lymph nodes from control and Krt17 KD 4T1 cells injected mice, show reduced Krt14 in Krt17 KD tumors and lymph nodes ($n = 2$/tumors with 6 FOV/tumor and $n = 7$ individual lymph nodes). **i** Graphical representation of tumor induction and progression in Rear mice injected with primary EpCAM[+] tumor cells from C3-T HET tumors. **j**, **k** Growth curve of tumors in Rear mice ($n = 4$ tumors/group, 2 independent biological experiments) injected with primary cells C3-T HET with Krt17 KD show reduced tumor progression in Krt17 KD cells. Representative IF images **l** and scatter blot **m** show reduced lymph node metastasis in Krt17 KD C3-HET tumor cells compared to control ($n = 5$ lymph nodes/group). Statistical significance was determined by two-way ANOVA with Tukey's multiple comparisons test in (**a**, **j**) *$P < 0.05$, **$P < 0.01$, ***$P \leq 0.001$, and ****$P \leq 0.0001$, and one-way ANOVA with Tukey's multiple comparisons test in (**d**, **g**, **h**, **m**); The data are presented as means ± SEM. **g**, **h**, **m** scoring was performed using indicated FOV and Scale bars, 100 μm.

Examination of the T-cells of control and Krt17 KD tumors show that the Krt17 KD 4T1 tumors showed an increase in the number of CD8[+] T-cells (Fig. 5g, h) and decreased the CD4[+] T-cells population (Fig. 5i, j) by IHC, which was further corroborated in FACS analysis (Supplementary fig. 3a). Further interrogation in CD4 and CD8 T-cells delineate a reduced number of CD8[−] CD4[−] T-cells in Krt17 KD 4T1 tumors compared to the control group by FACS analysis (Supplementary Fig. 3a). Recently growing research evidence

indicates that γδ T-cells play a dual role in cancer progression[47,48]. In contrast to other T-cells, γδ T-cells do not express CD8 and CD4[49]. Based on this data, we further examined the γδ T-cells population in the CD8[−] CD4[−] T-cells of the Krt17 KD tumors. Interestingly, the Krt17 KD 4T1 tumors showed significantly reduced number of γδ T-cells population as compared to the control (Fig. 5k, l). We found similar results in another TNBC model, where EpCAM[+] C3-T HET[35,36] primary TNBC tumor cells were knocked down for Krt17

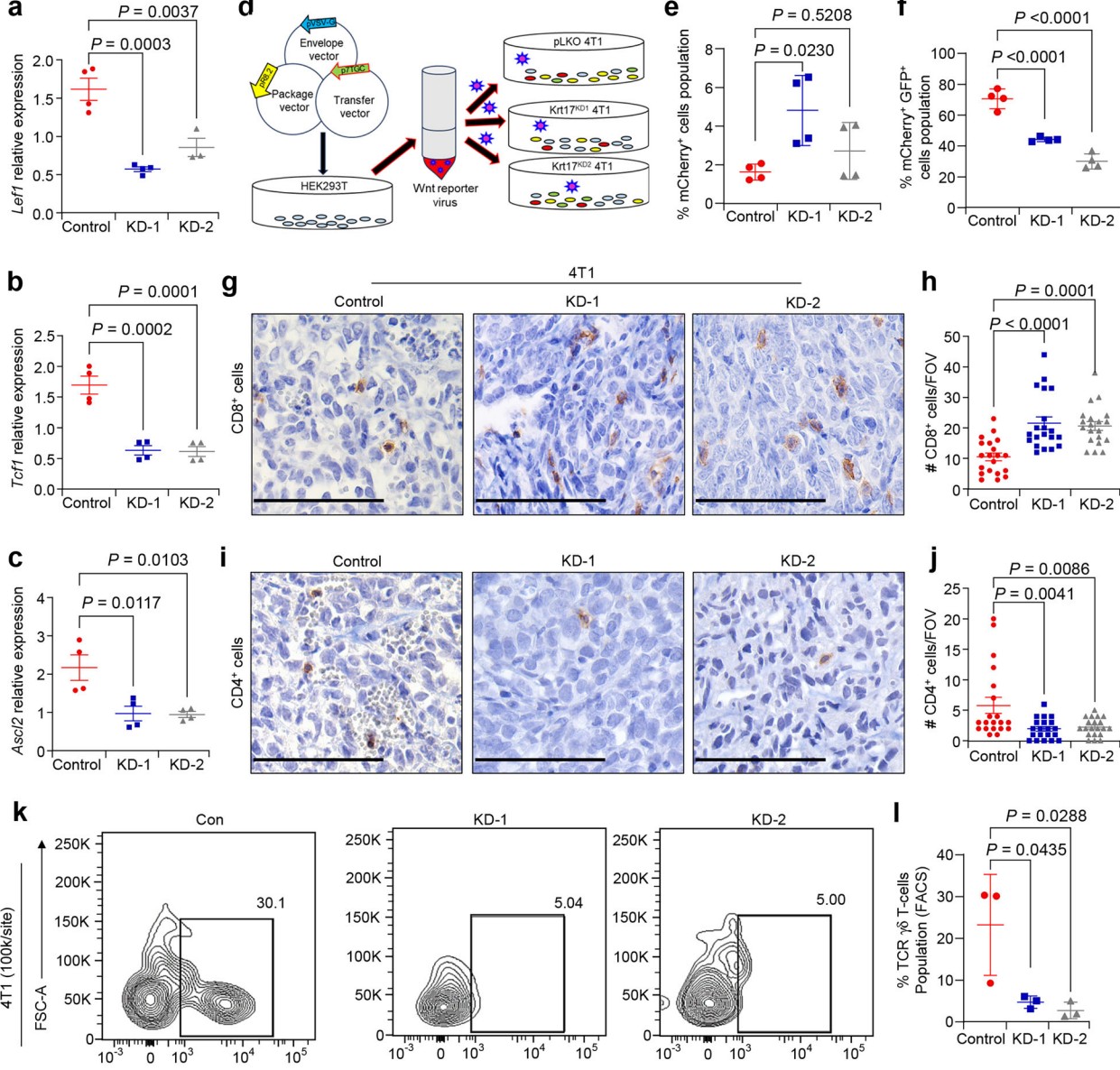

**Fig. 5 | Krt17 KD regulates Wnt signaling and modulates theT-cells levels in TNBC.** qRT-PCR analyses show reduced levels of Wnt target genes, **a** *Lef1* **b** *Tcf1* **c** *Ascl2* in Krt17 KD 4T1 cells compared to control. Two biological samples were used with technical duplicates, target genes expression values were normalized with *Gapdh*. **d** Schematic representation of Wnt reporter assay in 4T1 cells. **e, f** Wnt activities were detected by FACS based reporter assay (mCherry⁺ GFP⁺), mCherry was a readout of infection efficiency (*n* = 4 samples/group). **g, h** Immunohistochemical analysis of cytotoxic CD8 T-cells show increased number in Krt17 KD 4T1 tumors (*n* = 4 individual tumors with 5 random FOV/individual tumors). **i, j** Immunohistochemical analysis of CD4 T-cells shows reduced number in Krt17 KD 4T1 tumors (*n* = 4 individual tumors with 5 random FOV/individual tumors). **k** FACS plots showing γδ T-cells in different tumors of indicated groups. **l** Scatter plot shows a reduced population of γδ T-cells in different tumors by FACS (*n* = 3 tumors/group). Data are presented as the mean ± SEM. Statistical significance was determined by one-way ANOVA with Tukey's multiple comparisons test. Scale bars, 100 μm.

using lentivirus[40], and control and Krt17 KD tumor cells were injected into MFP of mice for tumor growth. Krt17 KD C3-T HET tumors showed significantly reduced number of CD4⁺ T-cells and concurrently increased number of CD8⁺ T-cells compared to control (Supplementary Fig. 3b, c), highlighting altered immune cells in TME in the absence of Krt17, supporting 4T1 tumor data (Fig. 5k, l). γδ T-cells were significantly reduced in Krt17 KD C3-T HET tumors compared to control tumors (Supplementary Fig. 3e, f and Supplementary Fig. 3h). CSCs also showed a modest reduction in number in Krt17 KD tumors compared to control tumors, suggesting a positive effect of Krt17 on CSCs (Supplementary Fig. 3g). These results indicate that Krt17 may regulate the number of γδ T-cells in TME

and activation of Wnt signaling potentiating CSCs in TNBC tumor cells.

**Pharmacological inhibition of the Wnt signaling delays tumor growth and metastasis through the suppression of γδ T-cells**
Next, we wanted to investigate if pharmacological blocking of Wnt signaling could reverse tumor promoting effects of Krt17. Thus, we assessed the inhibition of Wnt signaling using LGK-974 in TNBC mouse model. EpCAM⁺ tumor cells from C3T-HET mouse tumor were injected into Rear mice. LGK-974 is a specific porcupine protein inhibitor[50], it affects post-translational palmitoylation, an essential step in the Wnt ligand secretion process. Also, this drug exhibited preclinical activity in numerous tumor

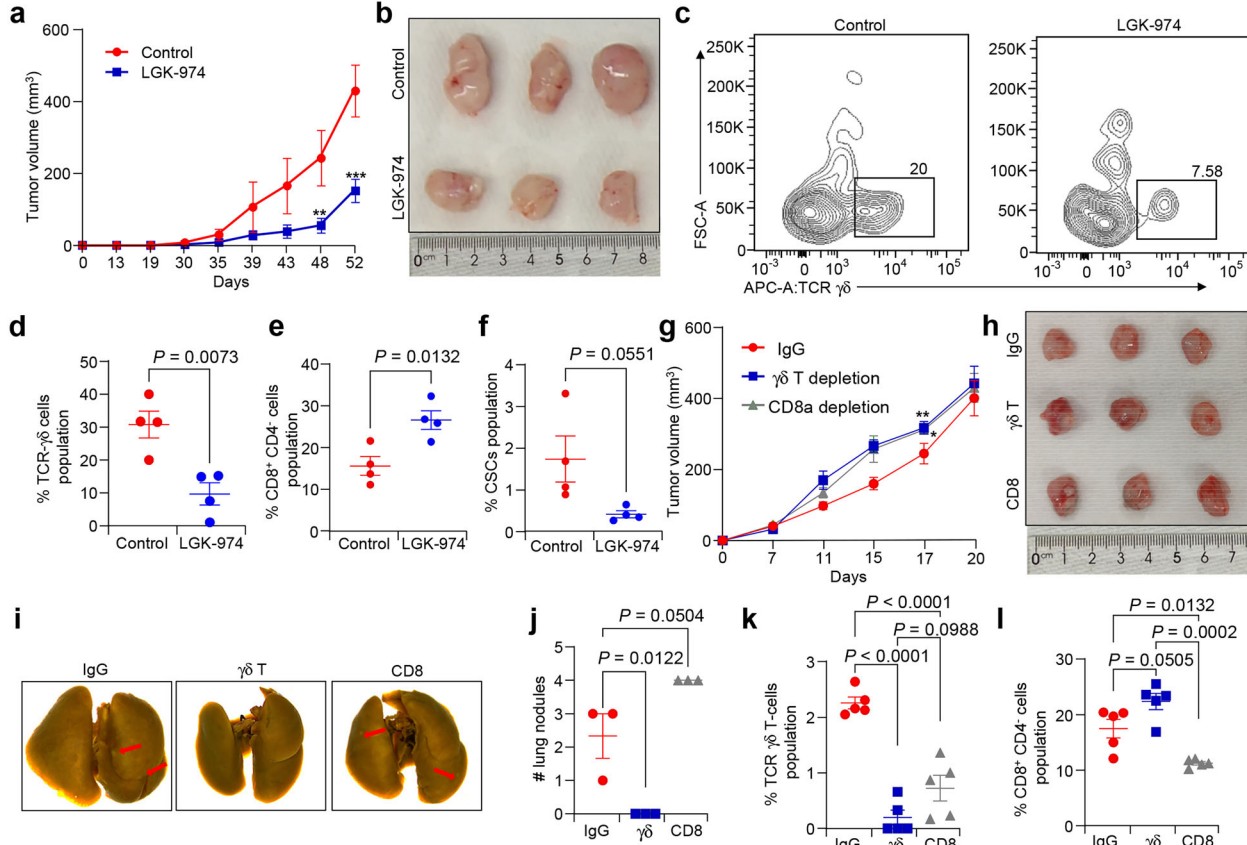

**Fig. 6 | Inhibition of Wnt signaling reduces γδ T-cells population in TNBC mouse models. a, b** Treatment with LGK-974 reduces tumor growth from EpCAM⁺ tumor cells from C3T-HET tumors compared to control (*n* = 4 tumors/group). **c,d** Reduced number of γδ T-cells in LGK-974 treated tumors compared to control tumors (n = 4 tumors/group). **e** Increased number of CD8⁺ T-cells population in LGK-974 treated tumors from C3T-HET EpCAM⁺ tumor cells compared to IgG control. **f** Number of CSCs significantly reduced in LGK-974 treated mice compared to control (*n* = 4 tumor/group). **g, h** Tumor progression of in 4T1 TNBC cell driven growth curve with control, blocking γδ and CD8 T-cells (*n* = 6 tumors/group). **i, j** Representative lung images show reduced number of lung metastasis nodules

with treatment of anti-TCR-γδ antibody compared to control. Lung metastasis nodules increased in tumor bearing mice treated with anti-CD8-T-cell antibody (*n* = 3 mice/group). The scattered plot shows the altered levels of γδ T-cells **k** and CD8⁺CD4⁻ **l** by FACS in tumors from mice with anti-TCR-γδ and anti-CD8 blocking antibodies, (*n* = 5 tumors/groups). Statistical significance was determined by two-way ANOVA with Tukey's multiple comparisons test in (**a, g**) *\**P* < 0.05, \*\**P* < 0.01, \*\*\**P* ≤ 0.001, and Student's t-test was used to determine p value in (**d, e, f**) and one-way ANOVA with Tukey's multiple comparisons test in (**j, k, l**); The data are presented as means ± SEM.

models, as well as currently it is being tested in clinical trial (Clinical-Trials.govidentifier: NCT01351103)[51]. Tumor growth was suppressed when mice were treated with LGK-974 at concentration of 5 mg/kg orally compared to saline control (Fig. 6a, b). Consistent with tumor growth decrease, we found increased CD8⁺ T-cells populations in the LGK-974 treated tumors and decreased γδ T-cells in LGK-974 treated tumors (Fig. 6c–e). Inhibition of Wnt signaling decreased the CSC population (Fig. 6f) in LGK-974 treated tumors, suggesting that these CSCs are most likely Wnt signaling responsive. These results confirmed that inhibition of Wnt signaling in TNBC impaired their tumor progression, metastasis possibly through suppression of γδ T-cells populations. These results suggest that the Wnt signaling plays a key role in CSC-mediated TNBC tumor progression by affecting γδ T-cells population.

To investigate the functional importance of γδ and CD8⁺ T-cells in the TME of Krt17 KD tumors, we performed MFP injection of 4T1 TNBC cells followed by blocking γδ T-cells and CD8⁺ T-cells by anti-T-cell receptor (TCR)− γδ and anti-CD8 antibodies, respectively. Blocking γδ T-cells suppressed lung metastasis, but there were no significant changes in the tumor size at 250 μg/mouse concentration. Blocking CD8⁺ T-cells at a concentration of 100 μg/mouse promoted lung metastasis without tumor size changes (Fig. 6g–j), highlighting inverse correlation of CD8⁺ T-cells with metastasis. We further analyzed γδ T-cells in tumors treated with γδ T depletion antibody and found significantly suppressed γδ T-cells population

showing efficacy of depletion antibody (Fig. 6k, *P* < 0.0001). Similarly, blocking CD8⁺ T-cells suppressed CD8⁺ T-cells population in tumors (Fig. 6l) as compared to the IgG control group mice, showing success of depletion antibody against CD8⁺ T-cells. Taken together, these results suggested that the γδ T-cells play an important role in metastasis in TNBC through CD8⁺ T-cells.

## Keratin 17 regulates Wnt signaling in BA TNBC patients along with increased γδ T-cells

To investigate the validity of the newly identified mechanism Krt17-Wnt signaling promoting CSC mediated tumor progression and metastasis in TNBC mice models into clinical samples, and also to discern their validity in disparate outcome in BA TNBC patients, we examined the activation of Wnt signaling in TNBC BA (*n* = 5) and WA (*n* = 5) patient tumor samples using qPCR. Interestingly, we observed that mRNA expression of *LEF1* and *LBH*, Wnt targets, significantly increased in BA tumor samples as compared to the WA tumor (Supplementary Table 5) (Fig. 7a, b). Activation of Wnt signaling was further confirmed in multiple patient derived xenograft (PDX) derived cells in culture (Supplementary Table 6), which shows increased β-CATENIN expression in 4 independent BA TNBC cells compared to 3 independent WA TNBC cells (Fig. 7c–f). Additionally, nuclear localization of β-CATENIN was observed in 3 BA TNBC cells, suggesting activation of β-catenin derived Wnt signaling (Fig. 7c–f and supplementary

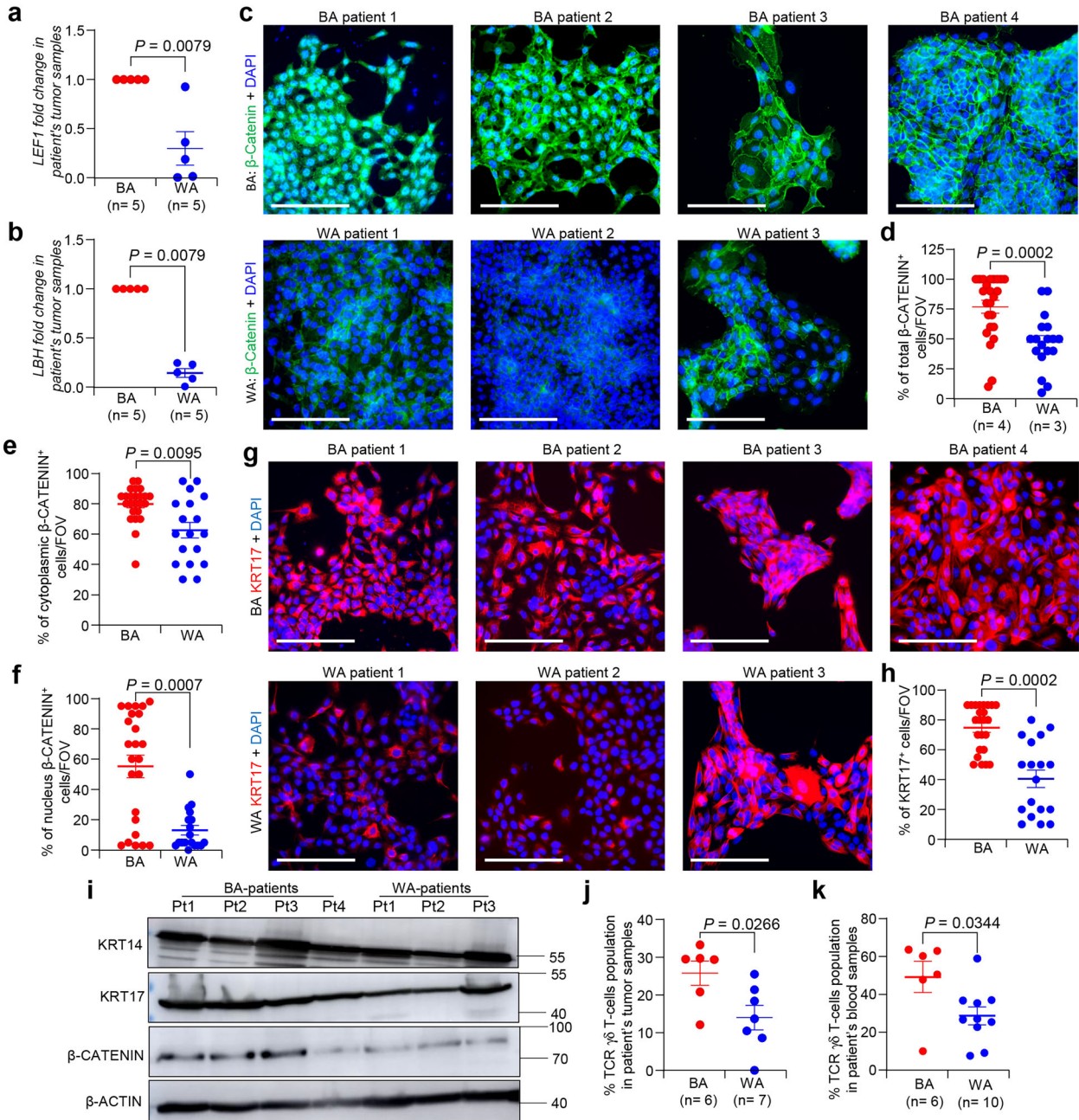

**Fig. 7 | BA TNBC patients has higher KRT17 and activated Wnt signaling and increased level of γδ T-cells. a, b** qRT-PCR data shows higher levels of Wnt target genes such as *LEF1* and *LBH* in BA compared to WA TNBC patient's tumor samples. Each dot represents each patient (*n* = 5 BA and *n* = 5 WA TNBC patients), and target genes expression values were normalized with *GAPDH*. **c–f** Immunofluorescence analysis shows higher β-catenin expression in 4 independent BA TNBC primary tumor cells compared to 3 independent WA TNBC patient's derived primary cells (*n* = 6 FOV/patient). Cytoplasmic and nuclear quantification of β-CATENIN is shown in **e, f**, respectively. **g, h** Immunofluorescence analysis shows higher KRT17 protein expression in 4 BA TNBC primary tumor cells compared to WA TNBC patient's derived primary cells (*n* = 6 FOV/patient). **i** Western blot analysis of KRT14, KRT17, and β-CATENIN in 4 BA and 3 WA TNBC patients derived monolayer cells (*n* = 2 individual experiments). **j, k** Scatter plot shows an increased number of γδ T-cells by FACS in TNBC BA (*n* = 6 BA and *n* = 7 WA tumors/group and *n* = 6 BA and *n* = 10 WA blood samples/group) compared to WA TNBC tumor and blood samples, respectively. Mann–Whitney *U* test was used to compute statistical significance in (**a, b, d, e, f, h**). Student's *t*-test was used to determine *p* value in (**j, k**). Data are presented as the mean ± SEM. Scale bars, 100 μm.

Fig. 4a, b). Notably, BA TNBC patient cells also showed higher KRT17 protein expression compared to WA TNBC cells (Fig. 7g, h) as demonstrated by immunofluorescence assay. These results were also confirmed by western blot of β-CATENIN and KRT17 in the PDX derived tumor cells from TNBC BA and WA patients (Fig. 7i). Also, western blot analysis showed that the expression of KRT14 was comparable between samples from Black and White American patients. Furthermore, immunofluorescence assays with PDX derived organoids (PDXOs) derived from

WA and BA patients, show increased abundance of TP63 (Delta N isoform), a stem like marker[52] in BA TNBC PDXOs compared to WA PDXOs along with increased expression of KRT17 (Supplementary Fig. 5a). Notably, KRT14[+]KRT19[+] bipotent tumor cells which represent stem like properties[53] and are associated with aggressive disease were also more in BA TNBC PDXOs (Supplementary Fig. 5b), suggesting more stem cell activity in BA TNBC cells compared to WA TNBC cells, supporting their connection to aggressive disease progression. To determine if BA TNBC have increased γδ

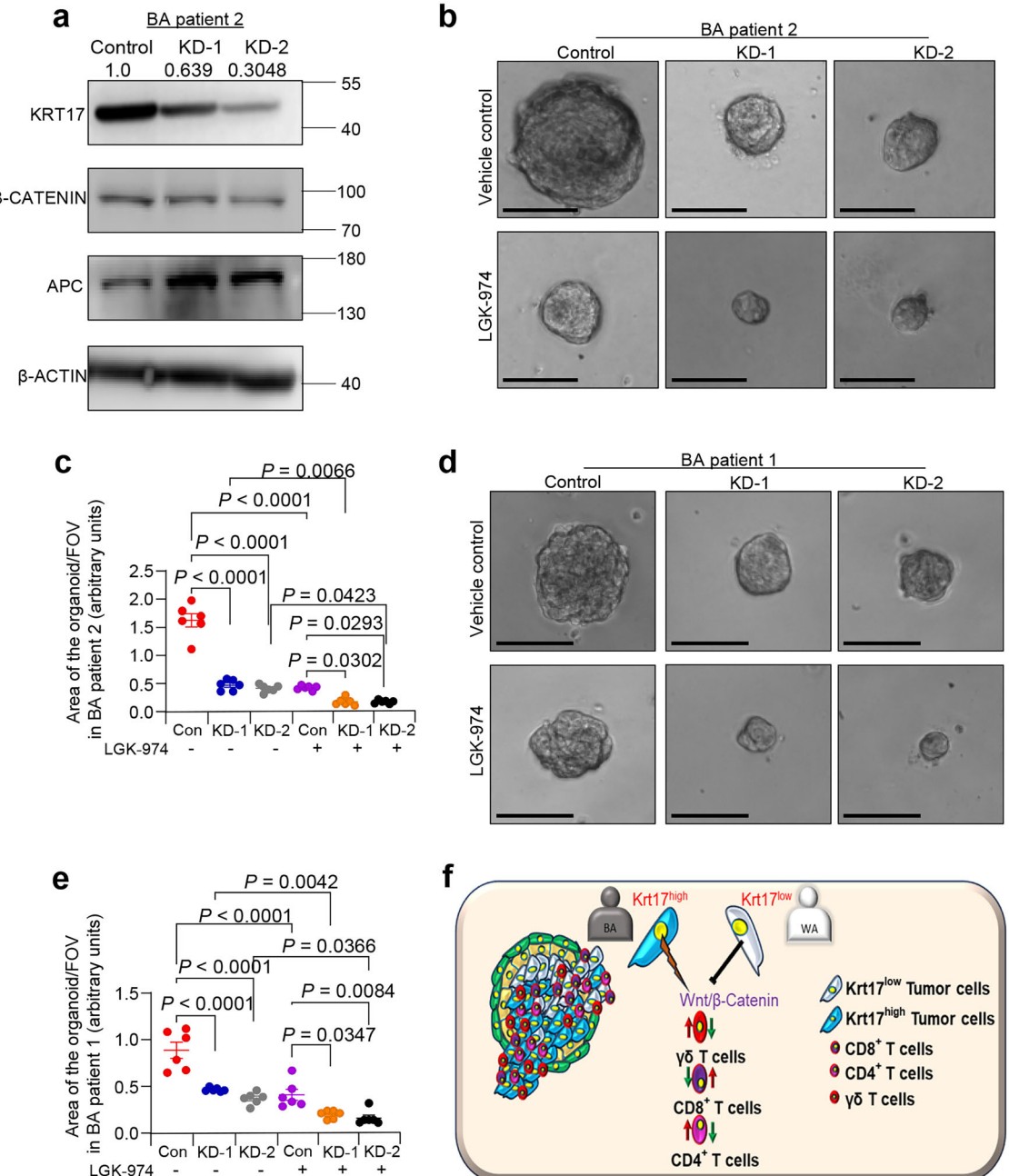

**Fig. 8 | Pharmacological blocking of Wnt signaling in Krt17$^{high}$ BA TNBC patients reduces CSCs dependent tumorsphere abilities. a** Representative western blot analysis KRT17, β-CATENIN and APC in KRT17-KD BSSR-2813 (BA TNBC) cells (*n* = 2 individual experiments). Band intensity of KRT17 was normalized to β-actin. **b–e** Representative tumorsphere images show reduced tumorsphere size in KRT17 KD BA TNBC cells (BSSR-2813 and 2402) compared to control. Addition of LGK-974 reduced tumorsphere size further in KRT17 KD cells (*n* = 6 FOV per group, 2 spheres per FOV were evaluated for area estimation, experiment repeated twice). **f** Graphical image depicts the *KRT17$^{high}$* tumor cells increased the activation of Wnt/β-catenin signaling and recruitment of γδ T-cells in their surrounding TME in BA TNBC patients compared to *KRT17$^{low}$* WA TNBC patients. Data are presented as the mean ± SEM. Statistical significance was determined by one-way ANOVA with Tukey's multiple comparisons test (**c**, **e**). Scale bars, 100 μm.

T-cells similar to mouse models (Supplementary Table 7), we performed FACS on fresh patient tissues obtained from iospecimen core, which shows a significant increase in the γδ T-cells populations in TNBC BA tumor samples (Fig. 7j, *n* = 13 patients) as well as in blood (Fig. 7k, *n* = 16 patients) samples from patients as compared to the WA TNBC patients by FACS analysis (Supplementary Table 8), suggesting recruitment of γδ T-cells populations in BA TNBC patients.

To investigate the causal function of KRT17 on Wnt signaling in human TNBC cells, we KD KRT17 in BA PDXO derived TNBC cells. Western blot confirms KD of KRT17 protein expression (Fig. 8a) in PDX derived monolayer cells (Based on band intensity, KD1 shows ~40% KD and KD2 shows ~70% KD). Notably, β-CATENIN expression is higher in BA TNBC cells. APC is an essential component of a protein complex that targets β-CATENIN for degradation[54]. So, we next examined if KRT17 regulates APC levels. APC protein level was increased in KRT17 KD BA cells, suggesting that KRT17 may downregulate APC to activate Wnt signaling through β-CATENIN (Fig. 8a). Organoid assay with control and KRT17 KD cells from two BA TNBC patients further demonstrates decrease organoid area in KRT17 KD group compared to control, which were further decreased by addition of LGK-974, a Wnt signaling inhibitor (Fig. 8b–e). We

performed similar experiments with established TNBC cell lines from ATCC (BA TNBC cells: HCC1806, MDA-MB-468, and WA TNBC cells: BT549 and HCC1937). BA TNBC cell lines have higher KRT17 expression than WA TNBC cell lines (Supplementary Fig. 6a), similar to PDX derived primary TNBC cells. We also observed increased expression of β-CATENIN in BA TNBC cells (Supplementary Fig. 6b) as compared to the WA cells, supporting earlier patient data. KD of KRT17 in HCC1806 and MDA-MB-468 (Supplementary Fig. 6c, d) BA TNBC cells, followed by tumorsphere assay, demonstrated that Krt17 KD suppressed sphere forming ability of BA TNBC cells (Supplementary Fig. 6e–h). Similar to TNBC PDX derived cells, we observed a similar decreased in the tumorsphere forming ability of BA TNBC cells in the presence of LGK-974, suggesting that effects of KRT17 are mediated through β-CATENIN dependent Wnt signaling (Supplementary Fig. 6e–h). Taken together, our results suggested that Krt17 is responsible for increased CSC function in tumor cells through β-catenin mediated activation of Wnt signaling in BA TNBC patients, and this is associated with pro-tumorigenic abundance of γδ T-cells in TME of BA TNBC patient tumors (Fig. 8f), which could be used as predictive manner to stratify patients for future targeted therapy against Wnt signaling.

## Discussion

Krt17 regulates cancer progression by numerous processes, including angiogenesis, invasion, metastasis, and stemness[55–58]. In basal cell skin carcinoma, over expression of Krt17 impacts the immune microenvironment and promotes cancer aggressiveness[59]. Of note, Krt17 acts as a key regulator of T-cells mediated inflammatory skin diseases[60]. However, the role and the function of Krt17 in a subtype specific manner in breast cancer remains poorly understood. In this report, we found that TNBC patients have higher KRT17 protein compared to non-TNBC patients, which is further corroborated in mouse TNBC and non-TNBC cells, followed by functional studies. We further demonstrate that TNBC patients have higher KRT17 protein expression and are connected to decreased survival. In particular, BA TNBC patients have a higher KRT17 expression in tumors compared to WA TNBC and non-TNBC patients, and KRT17 high BA TNBC patients have worse clinical outcomes than WA TNBC patients. The molecular mechanism responsible for poor clinical outcomes in BA TNBC patients remains elusive. Our studies show that Krt17 high expression could be used to predict patients with poor clinical outcomes, particularly in BA TNBC. Additionally, scRNA-seq and GSEA study revealed an increase in the CSCs and Wnt signaling pathway in Krt17+ tumor cells from aggressive TNBC mouse models. Our findings in TNBC corroborate with other cancers such as cervical cancer, where authors showed that cervical stem cells expressed high levels of the Krt17[61,62]. Similarly, studies from Jang et al (2022) and Ortiz Sanchez et al (2016) indicated that the Krt17 expressed in the oral and cervical spheres, which have stemlike properties[58,63].

The KD of Krt17 reduced tumor progression in TNBC subtype in animal preclinical models, and in BA TNBC patient derived organoids. The function of Krt17 is dependent on the activation of Wnt signaling mediated CSCs and increased level of γδ T-cells population in the TME. In support, we found that the inhibition of Wnt signaling decreased CSCs and γδ T-cells population in the TME, resulting in the slower progression of tumor. In particular, we found that KRT17 regulates both the level and localization of β-CATENIN in these tumors. These findings underscore the potential role of Krt17 in TNBC through activation of Wnt signaling. The Wnt signaling pathway is involved in a series of physiological and pathological processes, such as mammary gland development, stem cell homeostasis, cancer development, and cancer drug resistance[45,64,65]. Our analysis of Krt17 function with Wnt pathway components demonstrated that Krt17 acts as a key upstream regulator of Wnt signaling pathway, which was enriched in aggressive TNBC tumors, as Wnt-related genes, including Lef1, Tcf1, and Ascl2, were decreased in Krt17 KD tumor cells.

Mounting evidence suggested that TME has remarkable influences on cancer cells survival and metastasis[66–69]. Our evaluation of the γδ T-cells in immune TME revealed a striking reduced number in Krt17 KD tumors with concomitant increased number of CD8+ T cells. Based on these findings,

further in vivo studies using Wnt inhibitor and/or γδ T-cells depletion studies have been conducted, which show that decrease in Wnt signaling is associated with decrease of γδ T-cells in TME. These results may also indicate that Wnt signaling is a key mediator for increasing the γδ T-cells population and its dependent metastasis events in TNBC patients. In support, depleting γδ T-cells also leads to decrease in metastasis but not tumor volume. This result was consistent with a previous research study on breast cancer, which showed that depletion of γδ T-cells reduced metastasis without affecting tumor volume[70].

Supporting close relevance of our murine model for mechanistic studies, an activation of Wnt signaling and higher number of γδ T-cells were observed in BA TNBC tumor samples with high KRT17. BA TNBC cells also show a higher abundance of KRT14+/KRT19+ bipotent cells, and TP63+ stem like cells in patient organoids. Further, the KD of Krt17 reduced the tumorsphere forming abilities in BA TNBC cells, indicating function of Krt17 in BA TNBC CSCs. This decrease in PDX derived cells was further enhanced with treatment of LGK-974, Wnt signaling inhibitor. Western blot showing increase in APC in KRT17 KD PDX derived cells further suggests that KRT17 may downregulate APC levels to allow activation of Wnt signaling in these patients. Our studies also highlight an intriguing potential connection between Wnt signaling and the γδ T-cells driven tumorigenesis and metastasis downstream of Krt17 in disparate outcomes of BA TNBC patients, which could be targetable with Wnt signaling inhibition using FDA approved drug, LGK-974. Future studies will delineate detailed mechanistic insight on regulation of KRT17 and γδ T-cells through Wnt signaling. Based on the current data, we hypothesize that KRT17 high tumor cells secrete Wnt ligands feeding to tumor cells leading to activation of Wnt targets, which could be reverted by LGK-974 (known porcupine inhibitor, which prevents secretion of Wnt ligands from cells); however, these observations need future explorations. LGK-974 is currently used in clinical trials in head and neck squamous cell cancer, melanoma, TNBC, etc. (Trial no: NCT01351103). Based on the initial trial, drug is well tolerated in patients, suggesting it is safe to use in patients. We believe that this drug is most likely safer in patients compared to Tankyrase inhibitors or Vantictumab. Tankyrase inhibitors induce stabilization of Axin1/2, which are key components of the Wnt/β-catenin destruction complex, ultimately leading to reduced Wnt signaling. However, clinical trials on reported tankyrase inhibitors have been severely limited by severe toxicity in the gastrointestinal (GI) tract[71,72]. Also, it is associated with increased bone loss in preclinical mouse model studies[73]. Similarly, Vantictumab, a human IgG2 antibody against FZD7, was discontinued due to bone-related safety[74]. Compared to these drugs, LGK-974 has shown to be not having any toxicity issues and well tolerated in patients[51]. Based on these data, LGK-974 seems to be a safer drug targeting Wnt signaling with well tolerance in patients.

In conclusion, our findings reveal that aggressive TNBC expressing higher Krt17 have high Wnt signaling, which further accounts for the accumulation of γδ T-cells in their surrounding microenvironment, leading to metastasis in TNBC. Subtype specific high expression of KRT17 in TNBC tumor cells suggests a specific function of tumor cells expressing high KRT17. High expression of Wnt signaling in TNBC tumor cells is often seen compared to non-TNBC tumor cells and is associated with poor clinical outcome[75,76]. Our studies suggest that one of the upstream regulators of Wnt signaling in TNBC is mediated by KRT17, which then is asssociated to poor clinical outcome in TNBC patients. Thus, high KRT17 could be used as a predictive marker to stratify patients with poor clinical outcomes in TNBC patients, particularly in BA TNBC patients, followed by personalized intervention of anti-Wnt signaling therapy, opening new drug vulnerabilities in these patients.

Study limitations include the relatively small number of patient samples available for in vitro functional analyses due to tissue availability. Although our studies show that KRT17 is higher in BA TNBC patients (n = 58 patients), leading to increased Wnt signaling and γδ T-cells recruitment, some limitations exist, such as the small sample sizes of patient samples for functional studies in vitro due to tissue limitations. Future studies with large patient cohort in prospective study involving tumor and

adjacent normal tissue is needed for validation of KRT17 as a predictive biomarker. Moreover, subtype specific function of KRT17 needs to be explored in larger patient population beyond the sample size of $n = 58$ patients, which is provided in the manuscript.

## Methods

### Human patient samples

TNBC and non-TNBC human breast cancer specimens and normal adjacent tissues were procured from The Eastern Division of the Cooperative Human Tissue Network (CHTN), the University of North Carolina, in collaboration with Qing Zhang (currently at UT Southwestern Medical Center) and Sylvester Comprehensive Cancer Center, University of Miami. Fresh de-identified TNBC patient primary tumor tissues and blood samples were used for flow cytometry analysis, primary cell establishment, and obtained from the Sylvester Comprehensive Cancer Center, University of Miami, in collaboration with Melinda M Boone (IRB 20060858), Biospecimen core. The research ethics committee of the University of Miami approved the study (permission number: IRB 20060858), and the patients gave written informed consent. Race was self reported by patients at the time of sample collection in the biospecimen core in Sylvester Hospital. Participants who self-identified as Black or African American were classified as "Black American" (BA), and those who self-identified as White were classified as "White American" (WA). Supplementary Tables 1–8 provided information of patient tumors and blood samples were used for this study. The KRT17 H-Score was determined by multiplication of intensity and abundance of cells. Intensity was scored in the range of 0–3, and abundance with a range of 0–100. KRT17 expression in tumor-adjacent normal breast tissue was assessed by evaluating myoepithelial and luminal epithelial cells separately. Myoepithelial cells normally express KRT17 and serve as internal positive controls. For luminal epithelial cells, KRT17 expression was scored by a pathologist based on the percentage of positive cells (>50% or <50%) and staining intensity (1 = weak, 2 = moderate, 3 = strong).

### Cell culture and reagents

Mouse breast cancer cells 4T1 (procured from Yibin Kang, Princeton University), and E0771 (ATCC), and human TNBC cells HCC1806, MDA-MB468, BT549, LM2, SKBR3, HCC1008, HCC1937, and HEK293T (ATCC) were used in these studies. 4T1, E0771, MDA-MB468, BT549, LM2, HCC1008, HCC1937, SKBR3, and HEK293T were grown in Dulbecco's modified Eagle's medium (Sigma Aldrich). The 4T1 cell medium with addition of 5 μg/ml of Insulin (Sigma Aldrich). HCC1806 was obtained from ATCC. HCC1806 was grown at additional of 1% of sodium pyruvate (Gibco), 1% glutamax (Gibco), 1% MEM non-essential amino acids (Gibco). All cells were maintained with 10% fetal bovine serum and 1% penicillin and streptomycin (Invitrogen) at 37°C with 5% $CO_2$ in a humidified cell culture incubator. Cell lines were routinely checked and confirmed to be mycoplasma-negative by PCR analysis (ATCC), and their authenticity was confirmed by short-tandem repeat profiling.

### Single cell isolation for FACS and primary culture maintenance from breast cancer patient samples

The protocol for the isolation of single cell isolation for flow cytometry and in vitro primary culture of breast cells was obtained from the previously published protocols[77,78]. Single cells were stained with a combination of antibodies (listed in supplementary Table 10) for 30 min at room temperature in the dark. For in vitro primary culture, the single cell pellets from TNBC PDX mice tumor (Supplementary Table 6), were suspended in the advanced DMEM media containing Neuregulin 1 (5 nM), R-Spondin 3 (250 ng/ml), FGF7 (5 ng/ml), FGF 10 (20 ng/ml), EGF (5 ng/ml), Noggin (100 ng/ml), A83-01 (500 nM), Y-27632 (5 μM), SB202190 (500 nM), N-Acetylcysteine (1.25 mM), Nicotinamide (5 mM), Primocin (50 μg/ml), B27 (1X), Glutamax (1X), and HEPES (1X). All primary cells were maintained at 37°C with 5% $CO_2$ in a humidified cell culture incubator. Culture medium was changed every 2–3 days.

### Stable cell line preparation

Lentivirus was made by using Lipofectamine 2000 (Invitrogen) in HEK293T cells with Krt17 plasmids, and packaging and envelop plasmids such as R8.2 and VsVg. Lentivirus containing medium was collected in different time intervals and filtered with 0.45 μM syringe filter followed by concentration using Lenti-X concentrator and stored at −80 °C. Stable cell lines were prepared by puromycin selection (0.5 μg/ml) and verified by qRT-PCR for Krt17 expression. Established stable cell lines were used for subsequent experiments.

For Krt17 KD in primary cells, EpCAM+ C3-T HET tumor cells were spin infected with Krt17 virus following published protocol[52]. For human PDX derived cells, similar spin infection method was used.

### Animal experiments

Animal housing and experimental procedures were approved by the Institutional Animal Care and Use Committee (IACUC) of the University of Miami. We have complied with all relevant ethical regulations for animal use. To generate the C3-T+ Elf5+/− FVBmice, Elf5+/− mice (C3-T HET) were mated with C3-T+ antigen bearing FVB mice (C3-T WT)[35]. Genotyping with specific primers for LacZ (allele of Elf5+/−) and C3-T was done to ascertain the phenotype. For mammary fat pad (MFP) injections, 5–6 week-old female Balb/c mice were administered with 4T1 cells with the established protocol. Five to six week-old female Rear mice were procured from the Jackson Laboratory, and their genotype was confirmed by a specific protocol. EpCAM+ tumor cells from C3-T HET were enriched using CD326 (EpCAM) microbeads mouse-kit followed by orthotopic injection in MFP of Rear mice. Tumor progression, volume, and metastatic nodules in the lungs were evaluated thereafter following standard IACUC protocol, using calliper weekly twice, protocol allows maximum tumor volume of $20 \times 20$ mm. All tumors in the study were below that maximum tumor limit. Anti-IgG (100 μg/mouse), anti-CD8a (100 μg/mouse) and anti-TCR-γδ (250 μg/mouse) antibodies (Supplementary Table 9) were given twice a week through IP after the tumor size reached 150 mm³ for the preventative study setup. LGK-974 (S7143, Selleckchem, 5 mg/kg body weight) was given orally in every alternate day, and 2% DMSO was used as a solvent and vehicle control for LGK-974. Tumors were palpated as denoted in the respective experimental figure. At the experimental endpoint, mice were euthanized by cervical dislocation for the collection of primary tumors and metastatic tissues, such as lung and lymph node, for subsequent analysis. Blood samples were obtained from the retro-orbital sinus prior to euthanasia from mice post anesthesia as per IACUC protocol.

### Clonogenic growth assay

The steadily transfected single suspended TNBC and Non-TNBC mouse cells ($5 \times 10^2$) were continuously cultured into the 6 well cell culture plate up to clear visible colonies formed. Meanwhile, the medium was refurbished every 2 days. Then, colonies were fixed with ice cold methanol and stained with crystal violet. Finally, the colonies were taken images.

### Tumorsphere forming assay

Control and Krt17 KD mouse and human breast cancer cells were seeded into the ultra-low attachment plates at a density of $5 \times 10^2$ cells/wells. Plates were cultured for 7 days under standard tumorsphere media (DMEM/F-12 supplemented with serum substitute 1× B27, 20 ng/ml hEGF, and 20 ng/ml basic FGF). Finally, images were taken, and spheroids were counted.

### PDXOs generation and maintenance

For organoid culture, the single cell pellets from TNBC PDX tumors (Supplementary table 6), were suspended in the advanced DMEM media containing Neuregulin 1 (5 nM), R-Spondin 3 (250 ng/ml), FGF7 (5 ng/ml), FGF 10 (20 ng/ml), EGF (5 ng/ml), Noggin (100 ng/ml), A83-01 (500 nM), Y-27632 (5 μM), SB202190 (500 nM), N-Acetylcysteine (1.25 mM), Nicotinamide (5 mM), Primocin (50 μg/ml), B27 (1X), Glutamax (1X), and HEPES (1X). For organoids, $1–2.5 \times 10^4$ cells were seeded into the growth factor reduced matrigel (Corning cat#354230) coated 24 well chamber and

allowed to grow up to 7 days. Organoids were maintained at 37°C with 5% $CO_2$ in a humidified cell culture incubator. Mediums were changed carefully twice a day.

For continuous organoid cultures, mature organoids were enzymatically dissociated using 0.25% trypsin–EDTA, mechanically disaggregated into single cells, and re-embedded in fresh growth factor-reduced Matrigel. The cells were then maintained under identical culture conditions for subsequent passages. Organoid area per field of view was measured using ImageJ software.

## Real time q-PCR

RNA mini kit (Invitrogen) was employed to extract the total RNA from cells. Then, the first strand cDNA was synthesized with Superscript IV kit (Invitrogen). Real-time PCR was performed using Quant Studio Design & Analysis Software v1.5.1 on a Quant Studio 3 Real-Time PCR Instrument (Applied biosystems). Gene expression was normalized with *Gapdh* (Respective primers are listed in supplementary Table 11).

## Immunohistochemistry, Immunofluorescence

The paraffin embedded tissues slices were deparaffinized and hydrated with gradient ethanol. After antigen retrieval, the section was blocked and incubated with the following primary antibodies for human tissues, KRT17 (1:50), and mouse tissues CD4 (1:25), CD8 (1:25), Krt14 (1:75) at 4°C overnight. For IHC secondary antibody was added to the slices at room temperature, and DAB chromogen was used for development of color. For IF detection of Krt14 expression, was done with fluorescent conjugated secondary antibody.

Immunofluorescence was carried out on mice and patient-derived cells on cover slips. After 80% confluency, cells were fixed with formalin for 20 min and washed thrice with 1X PBS. Then, coverslips were blocked at RT for 1 h with 0.1% Triton X-100 containing goat serum. After blocking, the residual blocking solutions were removed and washed with 1X PBS. Primary antibodies were diluted at 1:100 (KRT17), 1:25 (β-CATENIN) for human primary cells and 1:50 (Krt17) for mouse cancer cells and incubated overnight at 4°C. Fluorescence conjugated secondary antibodies and DAPI were applied. Immunohistochemistry and immunofluorescence images were acquired using a Leica microscope.

For mouse tumor tissues, we manually counted % or number of the cells with positive staining. For CD4/CD8 scoring, number of positive cells was quantified. Also, we manually counted % of the fluorescence positive cells by counting positive cells and normalizing the number to total cells, as seen by DAPI-positive cells.

For organoids, 7days-old 8-well chamber organoids were fixed with 2% formalin for 20 min at room temperature. After 20 min, organoids were permeabilized with 0.5% Triton-X 100 (1X PBS) for 10 min at 4°C. Then, permeabilized organoids were washed thrice with 1X PBS-Glycine buffer (130 mM NaCl, 13 mM Na2HPO4, 3.4 mM NaH2PO4, and 100 mM glycine) for 10 min each. After washing, the organoids were blocked with 10% goat serum for 60 min. Primary antibodies KRT17, TP63, KRT14, and KRT19 were used at different dilutions (Supplementary Table 9) and incubated overnight at 4°C. After primary antibody incubation, the chamber slide was allowed to sit at room temperature for 30 min (to harden the matrigel and easy to wash). The organoids were washed 3 times with IF wash buffer (130 mM NaCl, 13 mM $Na_2HPO_4$, 3.4 mM $NaH_2PO_4$, 7.7 mM $NaN_3$, 0.05% BSA, 0.2% Triton-X 100, and 0.04% Tween-20- pH 7.4) for 20 min. Organoids were incubated with secondary antibodies at room temperature for 60 min. Then, washed one time with IF wash buffer and twice with 1X PBS. Finally, the organoids were incubated with DAPI (1:3000) and mounted with Vectashield antifade mounting medium. The confocal images were captured using Olympus Fluoview FV3000 microscope.

## Western blot

The human cell protein lysates were resolved on SDS-PAGE gel, and the proteins were transferred to PVDF membrane (Millipore). After transfer, the membrane was blocked with 5% nonfat dry milk (Blotting-Grade Blocker-Biorad), and then incubated overnight at 4°C with different antibody diluted at 1:1000 in TBST. The next day, the membranes were incubated with an anti-mouse or rabbit HRP-linked antibody for 1 h and developed using ECL (Thermo Scientific). The density of bands was determined by ImageJ program, normalized to the level of β-ACTIN.

## Flow cytometry based Wnt reporter assay

The puromycin selected control, and KD of Krt17 cells were again transfected with fluorescent based reporter 7TGC (#24304, Addgene)[35,46] lentivirus and cultured with complete DMEM and supplemented with insulin. Afterward, the cultured cells expressing the fluorescent proteins of GFP and mCherry were detected by LSR Fortessa and analyzed with flow zone. The mCherry fluorescence signal was used to measure infection efficiency, while the Wnt reporter was measured by GFP color.

## Statistics and reproducibility

The results were analyzed by GraphPad Prism (version 9). The tumor growth curve data were analyzed by the two-way analysis of variance (ANOVA) to compute statistical significance. For multiple comparisons, a one-way ANOVA with Sidak's and Tukey's multiple comparisons test was used. The significance of differences was calculated by two-tailed Student's *t*-test for normally distributed datasets with confidence intervals, and Mann–Whitney *U* test was used to compute statistical significance for samples with low "*n*". All experiments were repeated *n* = 2 or 3 times. For IHC and IF, scoring was carried out with multiple fields of view (FOV) per sample. Sample sizes for each experiment are specified in the figures and figure legends.

## Reporting summary

Further information on research design is available in the Nature Portfolio Reporting Summary linked to this article.

## Data availability

All relevant data are provided in the article or supplementary information. The numerical source data for the graphs in this study are provided in Supplementary Data. scRNA-seq data are available in the NCBI Sequence Read Archive (accession PRJNA685201).

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

## Acknowledgements

We thank Biospecimen Shared Resource (BSSR, SCR_022889) for fresh deidentified breast cancer samples, Flow Cytometry Shared Resource (FCSR, SCR_022501) for cell sorting and FACS analysis and Onco-Genomics Shared Resource (OGSR: SCR_022502). These cores are part of the Sylvester Comprehensive Cancer Center at the University of Miami, which is supported by the National Cancer Institute (NCI) of the National Institutes of Health (NIH) under award number P30CA240139. This work was supported by grants from the American Cancer Society (RSG DDC-133604 and CSCC-Team-23-978452-01-CSCC), and National Cancer Institute (R01CA237243) to R.C. We also thank Eastern Division of the Cooperative Human Tissue Network (CHTN), University of Pennsylvania, and University of North Carolina, Chapel Hill, for providing de-identified human breast cancer fixed FFPE tissues. We thank Dr. Shanta Dhar and Shrita Sarkar for their help with confocal imaging of the patient tissue derived organoids. This work was supported by grants from NCI-R01 (R01CA237243) grant to R.C. and CSCC-ACS (CSCC-Team-23-978452-01-CSCC) and Breast Cancer Alliance grant and Breast Cancer Research Foundation (NXTGN-25-004) grants to R.C. This work was also financially supported by the NIH/NCI grant R01CA248158-01 (C.O.D.S.), NIH/NIA grant R01 AG069727-01 (C.O.D.S.), and the NIH/NCI 1R01CA284630 (C.O.D.S.).

## Author contributions

R.C. conceptualized the study and designed all experiments. C.P.S. performed most of the experiments, collection and analysis of data. G.T. performed mice work on LGK-974 treatment. U.K helped with all flow cytometry experiments. Y.T. analyzed the IHC slides in patients' samples. S.H. and C.O.D.S provided scRNA-seq data analysis. M.M.B. provided patient samples from Biospecimen core in University of Miami. R.C. wrote the manuscript with C.P.S, with feedback from all authors.

## Competing interests

The authors declare no competing interests.
