## [Transparent Peer Review file · Communications Biology]

KRT17 promotes triple negative breast cancer through activation of Wnt signaling and $\gamma\delta$ T cells recruitment

Corresponding Author: Dr Rumela Chakrabarti

Version 0:

Reviewer comments:

Reviewer #1

(Remarks to the Author)

The manuscript titled "KRT17 promotes TNBC progression through the activation of Wnt signaling and recruitment of $\gamma\delta$ T-cells" is well-written, methodologically sound, and provides significant advancements in understanding TNBC biology, particularly in the context of racial disparities. The role of KRT17 in cancer biology is well-introduced, setting the stage for the study's aims. The methods are detailed and well-documented and could be replicated by a skilled team. The figures are well-prepared, organized and clear. Key findings are visually emphasized using appropriate graphical formats (violin plots, scatter plots, fluorescence microscopy images etc) and the color schema effectively distinguish between groups. The figure legends provide detailed explanations for each figure, including experimental groups, methods, and statistical analyses. I do not see any obvious signs of intentional figure manipulation or duplications.

The major findings of the study are:

1. KRT17 is a biomarker for aggressive TNBC and poor prognosis (high KRT17 expression is associated with aggressive tumor characteristics, poor prognosis, and decreased distant metastasis-free survival (DMFS) in TNBC patients, and African American TNBC patients show higher KRT17 expression compared to white American TNBC patients and exhibit worse clinical outcomes).
2. KRT17 promotes TNBC progression via activation of Wnt signaling (KRT17 expression in TNBC tumors correlates with increased activation of β -catenin-dependent Wnt signaling, which is known to drive cancer progression and metastasis, and inhibition of KRT17 reduces Wnt signaling activity, resulting in diminished cancer stem cell (CSC) properties, which are critical for tumor aggressiveness and metastasis).
3. KRT17 drives immune modulation in the TNBC tumor microenvironment to promote TNBC progression (KRT17 expression increases the recruitment of pro-tumorigenic $\gamma\delta$ T-cells in the tumor microenvironment (TME), contributing to cancer progression, and KRT17 knockdown reduces $\gamma\delta$ T-cell populations and increases CD8+ cytotoxic T-cell populations, suggesting a shift toward an anti-tumor immune profile).
4. Importantly, the authors validate the findings in preclinical and clinical models (mouse models demonstrate that KRT17 knockdown reduces tumor growth, CSC populations, and metastasis, particularly to the lungs and lymph nodes, and patient-derived tumor samples confirm that African American TNBC tumors have higher levels of Wnt signaling activation and $\gamma\delta$ T-cell infiltration compared to white American TNBC tumors).
5. Furthermore, pharmacological inhibition of Wnt signaling using LGK-974 in TNBC mouse models suppresses tumor growth, reduces $\gamma\delta$ T-cells, and decreases CSC populations, which suggests that targeting Wnt signaling could be a viable therapeutic strategy, for TNBC patients with high KRT17 expression.
6. Lastly, the study identifies KRT17 as a potential PREDICTIVE biomarker for aggressive TNBC and poor prognosis in BA patients, supporting its role in stratifying patients for targeted therapy.

The findings are robust and supported by comprehensive data including animal models, single-cell RNA sequencing, functional assays, and patient-derived data.

This study would be a valuable addition to the field after addressing the following concerns:

1. Could there be alternative mechanisms linking KRT17 to TNBC aggressiveness? Please discuss.
2. How feasible is the transition from preclinical findings to patient treatment, especially concerning Wnt signaling inhibitors? Authors briefly mention the clinical trial for LGK-974 and FDA approval but a brief discussion of study findings (safety and

efficacy would strengthen their case). Also adding a discussion on tankyrase inhibitors and Vantictumab would provide a fuller picture of clinical applicability.

3. While the study addresses KRT17's role in TNBC in Black women, an analysis (even if in silico) and discussion of KRT17 on other racial/ethnic groups would provide a broader perspective.

The KRT17 KO mouse models are fantastic but did the authors consider PDX-models of KRT17 high vs low expressing tumors from patients. If so, a brief discussion would help justify methods and translational relevance of the findings of this study in the absence of more human clinical trial data.

4. Suggestions for figures: standardization of image scale bars would improve some figures (Supplementary Figures 3a–c). Supplementary Table-linked figures more explicitly linked to its corresponding tables/dataset would improve navigation and verification for the reader. In some cases, legends are lengthy and could benefit from brevity while maintaining essential details.

Reviewer #2

(Remarks to the Author)

The manuscript investigates the role of KRT17 in racial disparities observed in white Americans and African Americans in TNBC, as well as the ability of KRT17 to activate the WNT Pathway and promote metastasis. KRT17 gene expression was elevated in TNBC compared to Luminal and HER positive subtypes and was associated with poor prognosis based on Kaplan Meier analyses. Krt17 promoted WNT Pathway activation and metastasis.

Comments:

1. The choice of Student's T-Test in Figure 1 is less than ideal, given the small sample size. A Fisher's exact test would be better.

2. The rationale for using E0771 should be more apparent (Luminal B) in the Figure 3 legend.

3. Microscopy in Figure 7 should be clearer. Ideally, the beta-catenin and DAPI panels should be depicted separately and also merged. Beta-catenin localization should be ascertained (nuclear, adherens junctions in cytoplasm, etc). Is CTNNB1 increased, or is beta-catenin protein stabilized (such as by WAVE3 or loss of APC/GS3K3beta)?

4. The differences in beta-catenin and Krt17 expression in Figures 7c-f are compelling, but the sample sizes are extremely small. More samples should be examined. Furthermore, the Student's T-test should not be used on this small sample size. This finding could have profound implications not only for TNBC but also for others where there are tremendous inherent health disparities. What is the endogenous beta-catenin and Krt17 protein expression in black versus white American breast tissue? Is this a predisposing factor or part of the observed divergent pathogenesis?

Reviewer #3

(Remarks to the Author)

The manuscript submitted by Selvam et al investigates the mechanistic basis of the difference in triple negative breast cancer (TNBC) cancer mortality between black American (BA) and White American (WA) women. They provide multiple lines of in vitro and in vivo evidence implicating increased Krt17 levels in the activation of Wnt signaling, promotion of CSCs, and increase in T-cell levels in the promotion of TNBC progression and metastasis. Although the sample sizes used for the comparison of BA and WA tumor physiology (Fig 1 and 7) are small, the data suggests Krt17 dysregulation is more pronounced in BA TNBC patients than in WA patients and thus may contribute to the disparate mortalities in these cohorts. The use of in vivo models supports the clinical relevance of the study findings. The data is of sufficient quality and interest level for publication in Communications Biology. Minor edits are recommended.

Minor comments:

1. It would be beneficial to show that Krt17 protein levels are higher in TNBC than in other BC subtypes or adjacent noncancerous breast tissue. Figure 1i and 1j appear to address one of these points. but what is meant by "non-TNBC" is unclear.

2. The difference between BA and WA tumor KRT17 protein levels is significant in the tissues tested (Fig 1n, $p=0.045$) but not in CPTAC samples ($p=0.0693$). A discussion on this is needed.

3. Showing whether E0771 luminal B BC cells have lower baseline KRT17 levels than TNBC 4T1 (figure 3) would be helpful. The increase in tumorosphere size in luminal B cells (Fig 3K) resulting from KRT17 knockdown is opposite of what is observed in TNBC 4T1 cells. The implications of this finding should be discussed.

4. The text describing Figures 5e and 5f are unclear. Clearly stating that panel 5e shows that transduction efficiency was higher in the KD-1 samples, while 5f shows that both KD-1 and KD-2 reduce the % of transduced cells exhibiting Wnt activation would be helpful.

5. The data in Figure 6 is interpreted as indicating that " T-cells play an important role in metastasis through CD8+ cells." The basis for this statement appears to be shown in Fig 6k in which anti-CD8 reduces the % of T-cells, while the anti-antibody has no effect on the CD8+ cell fraction (Fig 6l). This point should be more clearly described for the reader.

6. Figures 7a and 7b show analysis of $n=2$ BA and $n=3$ WA tumor samples run in technical duplicate with all measurements shown as data points in the graphs. Plotting the averages of the technical replicates only and performing the t-test on the $n=2$ BA and $n=3$ WA values is more appropriate. The same applies for Fig 7d and 7f.

7. Including blots for TP63, KRT14, KRT19 and KRT17 in Supp. Fig 3E would strengthen the organoid IF data presented.

8. The model shown in Fig. 7m appears to imply WA tumors are Krt17-. Although BA tumors have higher Krt17 levels than WA, the data provided doesn't clearly show WA tumors are Krt17-. This could be addressed with added controls in Fig 1n.

Reviewer comments:

Reviewer #1

(Remarks to the Author)

I have carefully reviewed the revised manuscript and the authors' responses to previous reviewer comments. The manuscript presents important findings regarding KRT17's role in triple-negative breast cancer (TNBC), particularly in the context of racial disparities between Black American (BA) and White American (WA) patients. The revisions have substantially strengthened the work.

Major Strengths:

Enhanced clinical relevance- The addition of Asian TNBC patient samples (n=105) provides valuable cross-ethnic comparative data, strengthening claims about KRT17's differential expression patterns.

Improved sample sizes- The increase in patient cohorts (BA and WA TNBC from n=31 to n=58 in Fig. 1) and PDX-derived cells (from n=4 to n=7) significantly improves statistical power.

Mechanistic clarity- The additional data demonstrating KRT17 regulation of APC/ β -catenin pathway (Fig. 8a) provides clearer mechanistic insight into Wnt signaling activation.

Translational discussion- The expanded discussion of LGK-974 clinical applicability versus alternative Wnt inhibitors (tankyrase inhibitors, Vantictumab) is helpful and well-referenced.

Improved figures- Some IHC images (Fig. 1h, 1j, 1l) could benefit from higher-magnification insets to show cellular detail but the confocal images in Supp. Fig. 5 are of excellent quality.

Areas Requiring Attention:

Statistical methodology (Fig. 7a-b, 7d, 7f)- While sample sizes have improved (n=5 BA, n=5 WA for qPCR; n=4 BA, n=3 WA for IF), the statistical approach remains concerning for small sample sizes. For qPCR data in Fig. 7a-b: The legend states "each dot represents each patient" but doesn't clarify if technical replicates were averaged before statistical testing.

My recommendation is to explicitly state that biological replicates (not technical replicates) were used as n for statistical tests, and consider Mann-Whitney U test instead of t-test given small sample sizes and potential non-normal distributions.

Normal tissue controls (Supp. Fig. 1a-b)- The authors added normal adjacent tissue (NAT) data, which is valuable.

However, n=3 per group is insufficient for robust conclusions about baseline racial differences. The statement "we were unable to quantify the difference" due to different expression patterns weakens conclusions about whether KRT17 differences are tumor-specific or constitutional.

My recommendation is to explicitly acknowledge this limitation in the discussion and to suggest future studies with larger normal tissue cohorts.

β -catenin localization quantification (Fig. 7c, Supp. Fig. 4)- The revised manuscript shows nuclear β -catenin in "3 out of 4 BA TNBC patients" versus cytoplasmic in WA patients. However, no quantitative analysis of nuclear vs. cytoplasmic ratios is provided. My recommendation is to provide quantification of nuclear/cytoplasmic β -catenin ratios with statistical comparison, or at a minimum, provide blinded scoring by multiple evaluators.

E0771 contradictory findings (Fig. 3k-m)- The increased tumorsphere size with KRT17 knockdown in luminal B cells (opposite to TNBC) is intriguing but inadequately discussed. The authors state "KRT17 may have opposing functions in TNBC vs non-TNBC tumor cells" but don't explore mechanisms. This finding could undermine the proposed model if KRT17 has context-dependent effects. I recommend expanding the discussion of potential mechanisms for subtype-specific effects.

Minor Concerns:

$\gamma\delta$ T-cell mechanistic connection- The connection between Wnt signaling and $\gamma\delta$ T-cell recruitment remains somewhat unclear. Fig. 6 shows LGK-974 reduces $\gamma\delta$ T-cells, but whether this is direct or indirect is unclear. Adding brief discussion of potential mechanisms (Wnt ligands as chemoattractants? Indirect through other TME cells?) could help.

KRT17 knockdown efficiency- Fig. 8a shows Western blot for KRT17 KD in patient-derived cells, but quantification of knockdown efficiency is not provided; please add it (% reduction vs. control).

Reproducibility- Most experiments indicate n=3-4 biological replicates, which is acceptable. Some key experiments (e.g., organoid assays) should state number of independent experiments more clearly. Please ensure all figure legends clearly state the number of biological replicates, the number of independent experiments, and the number of technical replicates, where relevant.

Discussion section- Please add a paragraph discussing limitations, particularly small sample sizes for some patient cohorts, a need for prospective validation of KRT17 as predictive biomarker, unknown mechanisms of subtype-specific KRT17 effects, whether findings extend to other racial/ethnic groups beyond those studied especially since the difference between BA and Asian was not significant (p=0.2610).

Methods section- Please clarify how normal tissue KRT17 expression was assessed, given different patterns. Describe blinding procedures for histological scoring. Specify software and methods for IF quantification.

Results section- Consider tempering some conclusions, particularly regarding Asian patient comparisons. More clearly

distinguish KRT17-high vs. KRT17-positive tumors.

My overall assessment is that this is a substantially improved manuscript that makes important contributions to understanding TNBC disparities and identifying targetable mechanisms. The addition of comparative ethnic data, increased sample sizes, and mechanistic details regarding APC/ β -catenin regulation significantly strengthen the work. The translational implications regarding LGK-974 are potentially important for clinical trial design.

Once the methodological concerns, particularly regarding statistical approaches for small sample sizes, quantification of β -catenin localization, and discussion of contradictory findings in non-TNBC models are revised, the manuscript would make a strong contribution to the field.

My final recommendation: Minor revisions required before acceptance.

Reviewer #2

(Remarks to the Author)

My concerns were adequately addressed.

Reviewer #3

(Remarks to the Author)

The authors have addressed all the concerns raised in the revised manuscript.

Version 2:

Reviewer comments:

Reviewer #1

(Remarks to the Author)

I have carefully reviewed the revised manuscript and the authors' detailed responses to previous comments. I am satisfied that my concerns have been appropriately and thoughtfully addressed. The authors have made substantial revisions that strengthen the rigor, clarity, and interpretability of the work, particularly with respect to statistical methodology, transparency of experimental design, and discussion of limitations. I appreciate the authors' careful attention to detail and their willingness to incorporate constructive feedback throughout the manuscript. Overall, the revisions reflect significant effort and have materially improved the quality and impact of the study.

We thank the reviewers for their insightful comments, which have helped to improve the quality of our manuscript. We have addressed all points raised by the reviewers and have revised the manuscript significantly. We hope these revisions are sufficient for the manuscript to be accepted for publication in *Communications Biology*.

Reviewer #1:

1. Could there be alternative mechanisms linking KRT17 to TNBC aggressiveness? Please discuss.

Response: This is an interesting question. Mechanistically, our data suggests that Krt17 regulates Wnt signaling dependent increase in tumor progression and metastases and APC downregulation is involved in this process (**Fig. 8a**). Krt17 could be used to predict patients with poor clinical outcomes and downstream Wnt signaling could be targeted through Wnt signaling inhibitor such as LGK-974 (**Fig. 6, fig. 8b-e and, Supplementary Fig. 6e-h**). We further found that decreased Krt17 levels are associated with increased CD8⁺T cells and decrease infiltration of $\gamma\delta$ T-cells, which are also known to have pro-tumor effects (**Fig. 5g-n**). It is possible that Krt17 may have alternative mechanisms such as promoting angiogenesis or other downstream effector molecules, but that is beyond the scope of the current manuscript.

2. How feasible is the transition from preclinical findings to patient treatment, especially concerning Wnt signaling inhibitors? Authors briefly mention the clinical trial for LGK-974 and FDA approval but a brief discussion of study findings (safety and efficacy would strengthen their case). Also adding a discussion on tankyrase inhibitors and Vantictumab would provide a fuller picture of clinical applicability.

Response: Our data shows that Krt17 positive tumors may benefit from targeting Wnt signaling using LGK-974, as increased Krt17 is associated with aggressive TNBC along with increased Wnt signaling (**Fig. 6 and Supplementary Fig. 5c-f**). As mentioned above, high Keratin 17 levels in tumors could be used to predict patients with poor clinical outcome and downstream Wnt signaling could be targeted through Wnt signaling inhibitor such as LGK-974, which is an FDA approved drug and is in clinical trials in head and neck squamous cell cancer, melanoma, TNBC etc (Trial no: NCT01351103). Based on the initial trial, drug is well tolerated in patients suggesting it is safe to use in patients. We believe that this drug is most likely safer in patients compared to Tankyrase inhibitors or Vantictumab. Tankyrase inhibitors induce stabilization of Axin1/2 which are key components of the Wnt/ β -catenin destruction complex, ultimately leading to reduced Wnt signaling, However, clinical trials on reported tankyrase inhibitors have been severely limited by severe toxicity in the gastrointestinal (GI) tract (PMID: 32523090, PMID: 26692561). Also, it is associated with increased bone loss in mouse models (PMID: 29055830).

Similarly, Vantictumab, a human IgG2 antibody against FZD7, was discontinued due to bone-related safety (Diamond J.R. et al., 2020, PMID: 32803633). Compared to these drugs, LGK-974 has shown to be not having any toxicity issues and well tolerated in patients PMID: 33941878. Based on these data, LGK-974 seems to be a safer drug targeting Wnt signaling with well tolerance in patients. We have added this part in the discussion part of the revised manuscript.

3. While the study addresses KRT17's role in TNBC in Black women, an analysis (even if in silico) and discussion of KRT17 on other racial/ethnic groups would provide a broader perspective.

Response: We thank the reviewer for this comment. We were fortunate to obtain TMA of Asian breast cancer patients (n=105) who have TNBC from Creative Bioarray (Cat No: BRTMA258) (**Supplementary Fig. 1c and d**). We performed IHC on the TMA followed by pathological evaluation by breast cancer pathologist in the team. We found that Asian women also have high Krt17 protein expression, but compared to BA TNBC patients, Krt17 protein expression is still lower in the tumor cells in Asian patients. The Asian TNBC patient's data is added in the revised manuscript (**Supplementary Fig. 1c and d**).

4. The KRT17 KO mouse models are fantastic but did the authors consider PDX-models of KRT17 high vs low expressing tumors from patients. If so, a brief discussion would help justify methods and translational relevance of the findings of this study in the absence of more human clinical trial data.

Response: We thank the reviewer for this comment. We have increased PDX derived monolayer cells from n=4 to n=7 in the revised manuscript (**Fig. 7c-g**). We have performed IF with KRT17 on these PDX derived monolayer culture and demonstrated that KRT17 is higher in BA TNBC patients compared to WA TNBC patients (**Fig. 7e and f**). This data was further corroborated in 7 patient samples by WB, where we further showed that BA patients have higher KRT17 than WA TNBC patient cells (**Fig. 7g**). WB and IF also demonstrated that β -CATENIN is higher in BA TNBC cells compared to WA TNBC cells, thus connecting the KRT17-Wnt signaling in BA TNBC patients (**Fig. 7c, d and g**).

5. Suggestions for figures: standardization of image scale bars would improve some figures (Supplementary Figures 3a–c). Supplementary Table-linked figures more explicitly linked to its corresponding tables/dataset would improve navigation and verification for the reader. In some cases, legends are lengthy and could benefit from brevity while maintaining essential details.

Response: **Supplementary image 3a-c** (current supplementary Fig. 4a and b, and 5a and b) are confocal images and scale bar is as per confocal machine. We have provided patient information in the

supplementary tables 6 as per individual figures in the revised manuscript. We have shortened the lengthy legends in the revised manuscript.

Reviewer #2:

- 1. The choice of Student's T-Test in Figure 1 is less than ideal, given the small sample size. A Fisher's exact test would be better.**

Response: We thank the reviewer for the advice. In the revised manuscript, we have increased the TNBC patients sample number from n=31 to n= 58, particularly in **Fig. 1l and m**. With this large patient sample number, we have performed T-test as typically Fisher Exact test is done when sample size is less than 5.

- 2. The rationale for using E0771 should be more apparent (Luminal B) in the Figure 3 legend.**

Response: We have made this point clear in the revised manuscript that E0771 cells represent luminal B subset of breast cancer (**Fig. 3h-k**).

3. Microscopy in Figure 7 should be clearer. Ideally, the beta-catenin and DAPI panels should be depicted separately and merged. Beta-catenin localization should be ascertained (nuclear, adherens junctions in cytoplasm, etc). Is CTNNB1 increased, or is beta-catenin protein stabilized (such as by WAVE3 or loss of APC/GS3K3beta)?

Response: As per the reviewer's advice, we separately arranged the immunofluorescence panels images in the revised manuscript (**Supplementary Fig. 4a and b**). Combined DAPI and β -CATENIN levels are shown in **Fig. 7c**. IF data shows that 3 out of 4 BA TNBC patients have nucleus localization of β -catenin compared to WA TNBC patients who have cytoplasm staining for β -CATENIN (**Fig. 7c and Supplementary Fig. 4a and b**). WB of patient cells from BA and WA TNBC cells further show increased β -CATENIN levels in 4 individual BA TNBC patient derived cells compared to 3 independent WA TNBC cells (**Fig. 7g**). Thus, our data suggests that BA TNBC patients have higher levels and altered localization of β -CATENIN in tumor cells.

4. The differences in beta-catenin and Krt17 expression in Figures 7c-f are compelling, but the sample sizes are extremely small. More samples should be examined. Furthermore, the Student's T-test should not be used on this small sample size. This finding could have profound implications not only for TNBC but also for others where there are tremendous inherent health disparities. What is the endogenous beta-

catenin and Krt17 protein expression in black versus white American breast tissue? Is this a predisposing factor or part of the observed divergent pathogenesis?

Response: As per the suggestion of the reviewer, we have increased the number of BA and WA patients from n=2 to n=4 for BA and from n=2 to n=3 per WA TNBC patients (**Fig. 7c-f**). It is true that high KRT17 levels could be used to predict poor response in additional cancers such as Head and neck squamous cell cancer (**PMID: 35621713**). While we could not obtain normal breast tissues from BA and WA women, we were able to analyze adjacent normal tissue (NAT) to tumor tissues from BA and WA TNBC patients and found that breast cancer has higher expression of KRT17 compared to normal adjacent tissue (**Supplementary fig. 1a and b**). Expression of KRT17 in adjacent normal tissues of BA and WA TNBC patients seems similar, however, in cancer tissues, BA TNBC patients have higher expression compared to WA TNBC patients, suggesting an increase in expression in cancer context.

Reviewer #3:

1. It would be beneficial to show that Krt17 protein levels are higher in TNBC than in other BC subtypes or adjacent noncancerous breast tissue. Figure 1i and 1j appear to address one of these points. but what is meant by “non-TNBC” is unclear.

Response: We agree with the critical point raised by the reviewer. As per the reviewer's comment in this revised manuscript, we included the normal tissues adjacent to tumor tissues (NAT) in **supplementary fig 1a and b**. Our data shows that compared to normal tissues, breast cancer tissue has higher expression of KRT17 protein. Non-TNBC formalin fixed patient slides include tissue slides from ER+/PR+/Her2- patients (**Fig 1h and i**).

2. The difference between BA and WA tumor KRT17 protein levels is significant in the tissues tested (Fig 1n, p=0.045) but not in CPTAC samples (p=0.0693). A discussion on this is needed.

Response: Based on the reviewer's suggestions, we have significantly increased the number of TNBC patient tissue slides in both BA (revised manuscript we have 29 samples) and WA TNBC patients (revised manuscript have 29 samples) in **Fig 1l and m**. It is possible that the small sample size in CPTAC platform of n=18 patients may be responsible for non-significant number (**Current supplementary Fig.1e**). It is also possible that total KRT17 expression was evaluated in CPTAC platform rather than only tumor cells expressing KRT17, which is done by our pathologist in UM.

3. Showing whether E0771 luminal B BC cells have lower baseline KRT17 levels than TNBC 4T1 (figure 3) would be helpful. The increase in tumorsphere size in luminal B cells (Fig 3K) resulting from KRT17 knockdown is opposite of what is observed in TNBC 4T1 cells. The implications of this finding should be discussed.

Response: **Figure 3b** and clearly show the difference of baseline KRT17 protein levels in TNBC vs non-TNBC cells. It is true that loss of KRT17 in E0771 luminal B breast cancer cells show no significant difference in tumorsphere number (**Fig 3l**) compared to 4T1 TNBC cells (**Fig 3e and f**), which shows a reduced number of spheres. However, the size of tumorspheres from E0771 KD cells seems higher than control cells (**Fig 3m**). This data is in support of patient data shown in (**Fig. 1b**). We believe that KRT17 may have opposing functions in TNBC vs non-TNBC tumor cells.

4. The text describing Figures 5e and 5f are unclear. Clearly stating that panel 5e shows that transduction efficiency was higher in the KD-1 samples, while 5f shows that both KD-1 and KD-2 reduce the % of transduced cells exhibiting Wnt activation would be helpful.

Response: We thank the reviewer for this point and have made necessary changes to the revised manuscript.

5. The data in Figure 6 is interpreted as indicating that “ $\gamma\delta$ T-cells play an important role in metastasis through CD8+ cells.” The basis for this statement appears to be shown in Fig 6k in which anti-CD8 reduces the % of $\gamma\delta$ T-cells, while the anti- $\gamma\delta$ antibody has no effect on the CD8+ cell fraction (Fig 6l). This point should be more clearly described for the reader.

Response: We agree with the reviewer’s point and accordingly have modified the revised manuscript text.

6. Figures 7a and 7b show analysis of n=2 BA and n=3 WA tumor samples run in technical duplicate with all measurements shown as data points in the graphs. Plotting the averages of the technical replicates only and performing the t-test on the n=2 BA and n=3 WA values is more appropriate. The same applies for Fig 7d and 7f.

Response: We thank the reviewer for this comment. In our updated manuscript **fig. 7a and b**, we conducted qPCR experiments with more samples in BA (n=5) and WA (n=5), and each dot denotes individual patient samples. Interestingly, we observed increased mRNA expression of *LEF1* ($P = 0.0033$) and *LBH* ($P < 0.0001$), in BA TNBC tumor samples as compared to the WA tumor (**Supplementary table 5**). Also, we increased sample size in **fig 7d and f** in the revised manuscript BA (n=4) and WA (n=3).

7. Including blots for TP63, KRT14, KRT19 and KRT17 in Supp. Fig 3E would strengthen the organoid IF data presented.

Response: We thank the reviewer for this comment. Although we were unable to perform organoids western blot experiments due to the limitations of protein concentration in the fewer organoid derived cells. In the revised manuscript, we included the western blot data for KRT14, KRT17, β -CATENIN from BA and WA TNBC PDX derived monolayer cells (**Fig 7g**). We noticed that the KRT17, β -CATENIN expression was higher in BA TNBC patient derived monolayer cells as compared to WA samples.

8. The model shown in Fig. 7m appears to imply WA tumors are Krt17⁻. Although BA tumors have higher Krt17 levels than WA, the data provided doesn't clearly show WA tumors are Krt17⁻. This could be addressed with added controls in Fig 1n.

Response: We thank the reviewer for this important point. TNBC patients were segregated as KRT17^{high} and KRT17^{low} based on IHC. WA TNBC tumors are not KRT17 negative. We have added better representative images in **Figure1I** to represent this point. KRT17⁺ and KRT17⁻ was applied in single cell RNA sequencing study using mouse TNBC tumors. We applied that cut off to determine downstream signaling in KRT17⁺ TNBC tumor cells. We have modified the model (**Current fig. 8f**) accordingly in the revised manuscript.

Reviewers' comments:

Reviewer #1 (Remarks to the Author):

I have carefully reviewed the revised manuscript and the authors' responses to previous reviewer comments. The manuscript presents important findings regarding KRT17's role in triple-negative breast cancer (TNBC), particularly in the context of racial disparities between Black American (BA) and White American (WA) patients. The revisions have substantially strengthened the work.

Areas Requiring Attention:

Statistical methodology (Fig. 7a-b, 7d, 7f)- While sample sizes have improved (n=5 BA, n=5 WA for qPCR; n=4 BA, n=3 WA for IF), the statistical approach remains concerning for small sample sizes. For qPCR data in Fig. 7a-b: The legend states "each dot represents each patient" but doesn't clarify if technical replicates were averaged before statistical testing. My recommendation is to explicitly state that biological replicates (not technical replicates) were used as n for statistical tests, and consider Mann-Whitney U test instead of t-test given small sample sizes and potential non-normal distributions.

Response: We thank the reviewer for this comment. Fresh TNBC patient sample is limiting and often the tissue size is small for sorting which requires larger piece of tissues from patients. Hence, for qPCR using human samples in Fig. 7a and b, we used n=5 patients/group with technical duplicates were averaged. Please see information on 10 patients (5 BA and 5 WA) in supplementary table 5. Each dot represents one patient, and these 5 values were used for statistical analyses. We would like to highlight that the results were based on multiple experiments besides qPCR. We have also performed IF and WB analysis (Fig. 7c and i) on monolayer cell culture from additional 7 independent patients, where we analyzed n=4 BA and n=3 WA TNBC patients. We have now added Mann-Whitney U test for the qPCR analysis and have revised the fig 7a and b. Similarly, other figure panels are also modified for proper statistical tests.

Normal tissue controls (Supp. Fig. 1a-b)- The authors added normal adjacent tissue (NAT) data, which is valuable. However, n=3 per group is insufficient for robust conclusions about baseline racial differences. The statement "we were unable to quantify the difference" due to different expression patterns weakens conclusions about whether KRT17 differences are tumor-specific or constitutional. My recommendation is to explicitly acknowledge this limitation in the discussion and to suggest future studies with larger normal tissue cohorts.

Response: We have added necessary part in the revised discussion, where we acknowledged this low sample size for normal adjacent tissues.

E0771 contradictory findings (Fig. 3k-m)- The increased tumorsphere size with KRT17 knockdown in luminal B cells (opposite to TNBC) is intriguing but inadequately discussed. The authors state "KRT17 may have opposing functions in TNBC vs non-TNBC tumor cells" but don't explore mechanisms. This finding could undermine the proposed model if KRT17 has context-dependent effects. I recommend expanding the discussion of potential mechanisms for subtype-specific effects.

Response: Subtype specific function of Krt17 is expected based on data in Fig 1 having clinical findings, where we found that TNBC patients stratified with high KRT17 levels, show poorer prognosis compared to ER+ luminal A/B patients. One of the highlights of the manuscript is the context dependent data of KRT17, which delineates the unique function of KRT17 in TNBC, we have added this part in the discussion.

Minor Concerns:

$\gamma\delta$ T-cell mechanistic connection- The connection between Wnt signaling and $\gamma\delta$ T-cell recruitment remains somewhat unclear. Fig. 6 shows LGK-974 reduces $\gamma\delta$ T-cells, but whether this is direct or indirect is unclear. Adding brief discussion of potential mechanisms (Wnt ligands as chemoattractants? Indirect through other TME cells?) could help.

Response: We agree with the reviewer. Most likely Wnt ligands are being secreted more from KRT17 high tumor cells, which then aids more recruitment of $\gamma\delta$ T-cells. However, further detailed mechanistic insight of

this part is beyond the scope of the current manuscript. We have added necessary discussion in the revised manuscript.

KRT17 knockdown efficiency- Fig. 8a shows Western blot for KRT17 KD in patient-derived cells, but quantification of knockdown efficiency is not provided; please add it (% reduction vs. control).

Response: We have added the intensity of the bands normalized to Gapdh in the revised manuscript.

Reproducibility- Most experiments indicate n=3-4 biological replicates, which is acceptable. Some key experiments (e.g., organoid assays) should state number of independent experiments more clearly. Please ensure all figure legends clearly state the number of biological replicates, the number of independent experiments, and the number of technical replicates, where relevant.

Response: We thank the reviewer for this comment. All experiments are repeated at least twice for reproducibility. For organoids, we have performed staining for confocal imaging in 2 different experiments with multiple TNBC patients. We have modified the manuscript text accordingly.

Discussion section- Please add a paragraph discussing limitations, particularly small sample sizes for some patient cohorts, a need for prospective validation of KRT17 as predictive biomarker, unknown mechanisms of subtype-specific KRT17 effects, whether findings extend to other racial/ethnic groups beyond those studied especially since the difference between BA and Asian was not significant ($p=0.2610$).

Response: We thank the reviewer for this comment. We have added study limitations in the discussion.

Methods section- Please clarify how normal tissue KRT17 expression was assessed, given different patterns. Describe blinding procedures for histological scoring. Specify software and methods for IF quantification.

Response: Pathologist Dr. Youley Tjendra (co-author) performed scoring for all human tissues. Pathologist was blinded to the race information of the analysis. The KRT17 H-Score was determined by multiplication of intensity and abundance of cells. Intensity was scored in the range of 0-3 and abundance with a range of 0-100. KRT17 expression in tumor-adjacent normal breast tissue was assessed by evaluating myoepithelial and luminal epithelial cells separately. Myoepithelial cells normally express KRT17 and served as internal positive controls. For luminal epithelial cells, KRT17 expression was scored by a pathologist based on the percentage of positive cells (>50% or <50%) and staining intensity (1 = weak, 2 = moderate, 3 = strong). We have added this part in the method section.

For mouse tumor tissues (Fig. 5 g and i), we manually counted % or number of the cells with positive staining. For CD4/CD8 scoring, number of positive cells were quantified. Also, we manually counted % of the fluorescence positive cells by counting positive cells and normalizing the number to total cells as seen by DAPI+ cells.

Results section- Consider tempering some conclusions, particularly regarding Asian patient comparisons. More clearly distinguish KRT17-high vs. KRT17-positive tumors.

Response: We have made the necessary changes in the revised manuscript. KMplotter (free online software) allows segregation of patients based on gene levels, where we stratified patients based on KRT17 mRNA levels. In mouse tumors, in scRNAseq analysis, cells were clustered based on KRT17 mRNA levels >0 as positive cells. This criterion was used for defining KRT17⁺ and KRT17⁻ tumor cell population for further analysis.

My overall assessment is that this is a substantially improved manuscript that makes important contributions to understanding TNBC disparities and identifying targetable mechanisms. The addition of comparative ethnic data, increased sample sizes, and mechanistic details regarding APC/ β -catenin regulation significantly strengthen the work. The translational implications regarding LGK-974 are potentially important for clinical trial design.

Response: We thank the reviewer for this comment and appreciating the translational significance of our work.

Reviewer #2 (Remarks to the Author): My concerns were adequately addressed.

Response: We thank the reviewer for his/her time to critically read the manuscript and thoughtful comments to improve the manuscript.

Reviewer #3 (Remarks to the Author): The authors have addressed all the concerns raised in the revised manuscript.

Response: We thank the reviewer for his/her time to critically read the manuscript and thoughtful comments to improve the manuscript.